# FORMALRX: Rectify and eXamine Semantic Failures in Autoformalization

Haocheng Wang [* 1 2]  Baiyu Huang [* 1]  Yingjia Wan [* 3]  Xiao Zhu [1]  Xiaoyang Liu [4]
Yinya Huang [† 5]  Zhijiang Guo [† 1 6]

## Abstract

The veracious semantic alignment in autoformalization is significant for formal mathematical reasoning. However, existing evaluations provide only opaque binary verdicts or scalar scores, offering no interpretable insight into where or why translations fail. This opacity severely limits both human understanding and automated system improvement. To bridge this gap, we introduce FORMALRX, a comprehensive diagnostic evaluation framework that transforms autoformalization assessment from black-box judgments into actionable feedback. At its core is SCI Error Taxonomy, a hierarchical classification scheme decomposing autoformalization errors into 28 distinct categories with strict priority ordering. Building on this taxonomy, FORMALRX provides four critical diagnostic capabilities: alignment verdicts, error categorization, error localization, and correction. We instantiate the framework with a diagnostic model FORMALRX-8B, trained on 56,287 NL–FL pairs with fine-grained diagnostic annotations, and release FORMALRX-Test as the first fine-grained diagnostic benchmark. FORMALRX-8B achieves F1-scores of 0.88 (verdict) and 0.71 (categorization), along with accuracies of 0.75 (localization) and 0.73 (correction), substantially outperforming both general-purpose LLMs and specialized baselines. By connecting evaluation with actionable insights, FORMALRX enables systematic diagnosis and improvement of autoformalization systems.

[1]The Hong Kong University of Science and Technology (Guangzhou), Guangzhou, China [2]Department of Computer Science, ETH Zürich, Zürich, Switzerland [3]University of California, Los Angeles, USA [4]School of Mathematical Sciences, Shanghai Jiao Tong University, Shanghai, China [5]ETH AI Center, ETH Zürich, Zürich, Switzerland [6]The Hong Kong University of Science and Technology, Hong Kong SAR, China. Correspondence to: Yinya Huang <yinya.huang@ai.ethz.ch>, Zhijiang Guo <zhijiangguo@hkust-gz.edu.cn>.

*Proceedings of the 43rd International Conference on Machine Learning*, Seoul, South Korea. PMLR 306, 2026. Copyright 2026 by the author(s).

## 1. Introduction

Formal mathematical reasoning has emerged as a new frontier in Artificial Intelligence, offering a path to rigorous and verifiable logical inference (Yang et al., 2024). Within this paradigm, proof assistants such as Lean (de Moura & Ullrich, 2021; de Moura et al., 2015) have become pivotal. By grounding informal reasoning in strict formal systems, *autoformalization* translates mathematical problems from natural language into formal statements (Wu et al., 2022), serving as a critical bridge to mitigate the reasoning unfaithfulness and hallucinations (Wang et al., 2025b) frequently observed in Large Language Models (LLMs). However, the development of effective metrics and systematic evaluation frameworks to characterize translation failures remains notably limited (Yang et al., 2024; Weng et al., 2025).

Existing approaches to autoformalization evaluation are fundamentally limited by a lack of interpretability, hindering systematic error diagnosis and model improvement (Table 1). Early work relied on syntactic validity checks (Leanprover Community, 2024) and surface-level metrics such as BLEU (Papineni et al., 2002) or compiler type-check feedback (Azerbayev et al., 2023). Subsequent evaluations introduced rule-based equivalence checks (Liu et al., 2025b; Jana et al., 2025) and structural metrics (Liu et al., 2025c) to compare autoformalizations against formal ground-truth references, achieving high precision but suffering from low recall, limited domain coverage, and reliance on formal annotations. More recent methods (Lu et al., 2025; Gao et al., 2025; Yu et al., 2025a) relax the need for formal ground truths, yet still reduce semantic correctness to binary verdicts. Overall, existing evaluators offer no interpretable signals about the cause, location, or correction suggestion of semantic errors, limiting their usefulness for human debugging, model training, and scalable autoformalization.

In this paper, we introduce FORMALRX to transform autoformalization assessment from opaque binary judgments into actionable diagnostic feedback (Figure 1). Our contributions are threefold.

**SCI Error Taxonomy.** At the core of FORMALRX lies the SCI Error Taxonomy (Semantic, Constraint, Implementation), a hierarchical classification scheme that decomposes the monolithic notion of misalignment into 28

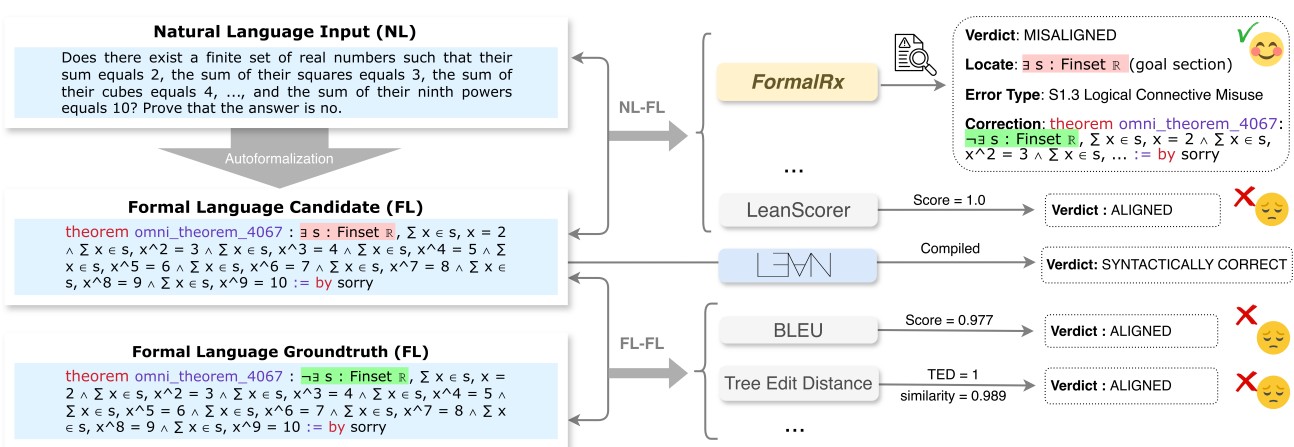

*Figure 1.* An illustrative example of FORMALRX compared with other semantic evaluation methods for autoformalization. On this misaligned candidate, all FL-FL and NL-FL baselines fail, scoring it as aligned or confirming only syntactic correctness, whereas FORMALRX returns the verdict, error category, error location, and a corrected statement.

distinct error categories with strict priority ordering to resolve ambiguity in error attribution. The taxonomy covers logical structure errors, mathematical object errors, constraint violations, and Lean 4-specific implementation errors, providing a principled and exhaustive basis for error diagnosis.

**FORMALRX Dataset & FORMALRX-Test.** Building upon the taxonomy, we construct a large-scale diagnostic dataset of annotated misaligned NL-FL pairs via taxonomy-guided error injection, each accompanied by full diagnostic metadata including error category, localization, and corrected statement. From this dataset, we hold out 7,030 samples as FORMALRX-Test, the first semantic evaluation benchmark for autoformalization with fine-grained diagnostic annotations beyond binary labels.

**FORMALRX-8B.** We train a diagnostic model on this dataset that jointly performs all four diagnostic tasks in a single forward pass: alignment verdict, error categorization, error localization, and correction. FORMALRX-8B achieves F1-scores of 0.88 (verdict) and 0.71 (categorization), along with accuracies of 0.75 (localization) and 0.73 (correction), substantially outperforming both general-purpose LLMs and specialized baselines.

## 2. Related Work

To address the limitations of automated compilers in verifying semantic fidelity, recent research has transitioned toward semantic evaluation methods for autoformalization. As shown in Table 1, these approaches are categorized into three streams based on the reference used for verification.

**FL-FL Comparison** compares the generated formal language (FL) statement against a ground-truth formal statement (FL). Initial methods (Wu et al., 2022; Azerbayev et al., 2023; Lu et al., 2024) utilized surface-form metrics like BLEU (Papineni et al., 2002) or basic type-checking, which often fail to capture deep semantic alignment. To improve precision, subsequent work introduced rule-based definitional equivalence (Liu et al., 2025b), domain-specific SMT solvers (Murphy et al., 2024), and structural operator-tree similarity (Liu et al., 2025d). While these advancements better handle syntactic variations, they remain limited by low recall in complex domains and an inherent dependency on the availability of gold-standard formalizations.

**NL-NL Comparison** utilizes a back-translation strategy, where the autoformalized output is translated back into natural language (NL) for comparison with the original natural language input (NL) (Ying et al., 2024; Gao et al., 2025; Yu et al., 2025b). This round-trip approach allows for evaluation without a formal ground truth; however, it frequently suffers from low recall. Semantic errors introduced during the initial autoformalization can be obscured during the back-translation process, allowing incorrect formal statements to appear consistent with the source text.

**NL-FL Comparison** directly assesses semantic consistency between natural-language statements (NL) and their formal counterparts (FL) without relying on reference formalizations. FormalAlign (Lu et al., 2025) uses contrastive and cross-entropy losses to learn semantic alignment. LLM-as-a-judge approaches employ majority voting (Ying et al., 2024) or unanimous agreement (Guo et al., 2025b) to aggregate judgments, with some methods decomposing statements into atomic subproblems or using multi-agent pipelines. However, existing NL–FL evaluation methods reduce semantic correctness to binary decisions with limited interpretability beyond scalar confidence scores. They provide no actionable feedback for correction, limiting their utility in scalable autoformalization pipelines. These limitations collectively motivate a shift from verdict-only evaluation to-

*Table 1.* Comparison of semantic evaluation methods for autoformalization. We categorize the diagnosis components as follows: ***Verdict***: the alignment decision on the autoformalization output; ***Categorize***: a fine-grained classification of the error type based on the proposed SCI error taxonomy; ***Locate***: the error location (if misaligned); ***Rectify***: the corrected formal statement output (if misaligned).

| Compa -rison | Source Paper | Type | Method | Compositional Diagnosis | | | |
| --- | --- | --- | --- | --- | --- | --- | --- |
| | | | | Verdict | Categorize | Locate | Rectify |
| FL-FL | ProofNet (Azerbayev et al., 2023) | Naive Matching | BLEU | Binary | ✗ | ✗ | ✗ |
| | ProofNet (Azerbayev et al., 2023) | Naive Matching | Typecheck | Binary | ✗ | ✗ | ✗ |
| | BEq (Liu et al., 2025b) | Rule-based | Equivalence Metric | Binary | ✗ | ✗ | ✗ |
| | ProofBridge (Jana et al., 2025) | LLM+Rule-based | Equivalence Metric | Binary | ✗ | ✗ | ✗ |
| | GTED (Liu et al., 2025d) | Rule-based | Tree Edit Distance | Binary | ✗ | ✗ | ✗ |
| | TransTED (Liu et al., 2025c) | Rule-based | Tree Edit Distance | Binary | ✗ | ✗ | ✗ |
| | LeanEuclid (Murphy et al., 2024) | Rule-based | SMT Solver | Binary | ✗ | ✗ | ✗ |
| NL-NL | LeanWorkbook (Ying et al., 2024) | Rule-based | NLI Test | Binary | ✗ | ✗ | ✗ |
| | Herald (Gao et al., 2025) | Rule-based | NLI Test | Binary | ✗ | ✗ | ✗ |
| | FormalMATH (Yu et al., 2025b) | LLM-as-Judge | Multi-LLM Ensemble Voting | Binary | ✗ | ✗ | ✗ |
| NL-FL | Lean Workbook (Ying et al., 2024) | LLM-as-Judge | Multi-LLM Majority Voting | Binary | ✗ | ✗ | ✗ |
| | ATF (Guo et al., 2025b) | LLM-as-Judge | Multi-LLM Ensemble Voting | Binary | ✗ | ✗ | ✗ |
| | ReForm (Chen et al., 2025) | LLM-as-Judge | Single-LLM | Binary | ✗ | ✗ | ✗ |
| | LeanScorer (Yu et al., 2025a) | LLM-as-Judge | Subtask Decomposition | Binary | ✗ | ✗ | ✗ |
| | AriaScorer (Wang et al., 2025a) | LLM-as-Judge | Subtask Decomposition | Binary | ✗ | ✗ | ✗ |
| | FormalAlign (Lu et al., 2025) | Training-based | SFT (CE+Contrastive Loss) | Binary | ✗ | ✗ | ✗ |
| | **FORMALRX (Ours)** | Training-based | SFT (CE Loss) | Multiple | ✓ | ✓ | ✓ |

ward interpretable, fine-grained diagnostic frameworks that can support both human debugging and automated model improvement.

## 3. SCI Error Taxonomy

To transform autoformalization assessment from opaque binary judgments into actionable diagnostic feedback, we propose **SCI Error Taxonomy** (Semantic, Implementation, and Constraint), a systematic three-tier taxonomy for formalization errors. The Implementation dimension is specific to the execution semantics of Lean 4 (de Moura & Ullrich, 2021; de Moura et al., 2015), while the Semantic and Constraint dimensions address fundamental mathematical formalization challenges that arise across different proof assistants. We anticipate that the core structure of the taxonomy could be adapted to systems such as Coq (Barras et al., 1997) and Isabelle (Nipkow et al., 2002), though this would require language-specific adjustments. Figure 2 presents the complete taxonomy with 28 error categories organized hierarchically. Complete category definitions with examples and edge cases are provided in Appendix F.1.

### 3.1. Design Principle

**Partition-Based Design.** Drawing from the definition of set partitions (Halmos, 1974; Rosen, 2011), we design the taxonomy to ensure error categories are pairwise disjoint and collectively exhaustive. Formally, let $\mathcal{E}$ be the set of all autoformalization errors. The taxonomy partitions $\mathcal{E}$ into three dimensions $\mathcal{P} = \{S, C, I\}$ such that for any two distinct dimensions $D_1, D_2 \in \mathcal{P}$, $D_1 \cap D_2 = \emptyset$, and

$\bigcup_{D \in \mathcal{P}} D = \mathcal{E}$. Each dimension is further partitioned into subcategories following the same principle.

**Priority Ordering.** In practice, certain misalignments may exhibit characteristics matching multiple categories. We resolve such ambiguities through hierarchical priority ordering. Specific error types take precedence over location-based classifications. For instance, when an error involves both a logical connective misuse (Figure 1) and occurs in a conclusion, the specific type classification takes priority over the location-based conclusion error (C4).

Within each category, subcategories with smaller numerical indices have higher priority when an error lacks clear categorization. This ordering principle, embedded in both the taxonomy design and its application during data synthesis (Section 4.3), guarantees that each error is assigned to exactly one category, achieving deterministic uniqueness in classification. To guarantee that all possible errors are captured, we incorporate location-based catch-all categories based on syntactic position within formal statements. Premise errors (C3) capture issues in statement assumptions, conclusion errors (C4) in statement goals, and auxiliary construction errors (C5) in supporting definitions. These categories are deliberately designed with lower priority than specific error types, functioning as fallback classifications for errors not fitting finer-grained categories. By partitioning the remaining error space through syntactic location, these catch-all categories ensure collective exhaustiveness. Any error not classified as a specific semantic, implementation, or constraint type can be unambiguously assigned based on where it occurs in the formal statement structure.

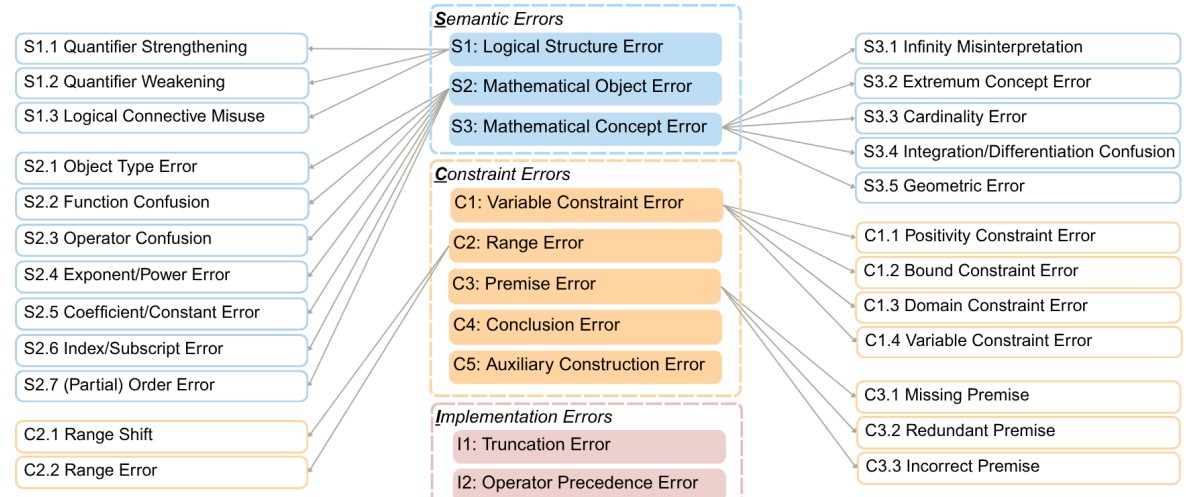

*Figure 2.* Overview of the SCI Taxonomy, which partitions 28 error categories into three disjoint, exhaustive dimensions: Semantic (S), Constraint (C), and Implementation (I) errors. Location-based catch-all categories ensure exhaustiveness, and priority ordering resolves cases that match more than one. See Section 3.3 for full definitions.

## 3.2. Development and Validation

**Iterative Empirical Development.** The SCI Error Taxonomy emerged through an iterative cycle of pattern identification, category abstraction, and empirical testing. We began by manually reviewing approximately 2,000 misaligned informal-formal statement pairs from both human-annotated formalizations and autoformalization outputs, identifying recurring error patterns (representative cases in Appendix G.1). These observations were iteratively abstracted into error-level categories by identifying common underlying causes, then organized into an initial three-dimensional structure. Category definitions were continuously refined through two validation strategies. First, from correct formalizations in Section 4.2 we used taxonomy-guided prompts (Appendix I.4) to generate negative samples where each ground-truth statement was modified to introduce specific error types (representative synthesis examples in Appendix G.2).

Manual inspection of generated samples assessed: (1) whether assigned categories accurately captured introduced errors (accuracy), and (2) whether all categories could be successfully instantiated (coverage). This validation informed iterative refinements to both category definitions and synthesis prompts. Second, we cross-referenced with existing formal-informal inconsistency datasets to identify potential gaps. This iterative refinement process continued until new instances rarely introduced distinct error types, indicating category saturation.

**Empirical Validation.** We validate the SCI Error Taxonomy through complementary analyses. First, large-scale synthesis yields 56,673 compiled error instances (Section 4), of which 52,521 receive stable and unambiguous category assignment after automatic re-tagging, indicating operational clarity of the taxonomy. Second, expert evaluation in Ap-

pendix C confirms human interpretability across four diagnostic tasks involving direct use of the taxonomy, notably the Validation and Re-tag stage where experts achieve 86.2% accuracy and 94.7% agreement, with most disagreements attributed to boundary ambiguities. Finally, a 200-sample manual spot-check (Appendix C.2) yields 98% correct labeling, supporting that taxonomy-guided synthesis produces consistent and reliable annotations.

## 3.3. Taxonomy Overview

**Semantic Errors.** This error captures failures in preserving mathematical content, rendering intended proofs either trivial or impossible. We organize semantic errors hierarchically by scope of impact. Logical structure errors occur at the propositional level when individual propositions are correctly formulated but mistakes arise in the logical relations connecting them, including over-strengthening or over-weakening statements, and errors in quantifiers and connectives. Mathematical object errors involve incorrect specification of objects within propositions where the underlying concept is understood but the object itself is misrepresented, such as confusing different types, functions, operators, or incorrectly specifying numerical parameters like exponents, coefficients, and indices. Mathematical concept errors stem from fundamental misunderstandings of concepts themselves, such as misinterpreting infinity, confusing bounds with extremum attainment, misrepresenting cardinality, confusing differentiation with integration relationships, and errors in geometric relationships.

**Constraint Errors.** This error addresses violations in specifying where and under what conditions mathematical statements hold. These errors include incorrect variable constraints that specify admissible value ranges, range errors in

iteration domains, premise errors in statement assumptions, conclusion errors in statement goals, and auxiliary construction errors in supporting definitions. Remaining categories ensure taxonomic completeness through structural decomposition based on syntactic location. As catch-all categories with lower priority than specific error types, premise errors occur in statement assumptions and can be missing, redundant, or incorrect, conclusion errors occur in statement goals, and auxiliary construction errors occur in helper definitions and lemmas outside the main statement. These location-based categories guarantee that any error not captured by finer-grained categories can be unambiguously classified.

**Implementation Errors.** Implementation Errors isolate discrepancies between mathematical notation conventions and formal language execution semantics. Unlike semantic errors that corrupt mathematical validity, implementation errors preserve the intended mathematical meaning but introduce computational discrepancies through language-specific behavior. Type-induced truncation arises when operations on discrete numeric types lead to unexpected results, including division truncation where division over natural numbers follows truncated semantics rather than real division, subtraction truncation where natural number subtraction automatically truncates to zero, and function output truncation where functions defined on discrete types return domain-specific values. A binding precedence mismatch occurs when the formal language parser assigns operator precedence that contradicts standard mathematical convention, causing expressions to be evaluated in unintended orders. These errors are easily overlooked during formalization as code compiles successfully. While Lean provides tactics to handle type conversions during proof construction (Doll et al., 2022), they cannot rectify errors already present in the statement itself. This dimension is inherently language-specific, capturing execution behaviors unique to Lean 4 that may differ in other proof assistants.

## 4. Dataset

Developing diagnostic capabilities requires large-scale training data with fine-grained annotations that extend beyond binary alignment judgments to encompass evaluation components described in Table 2. Existing datasets offer high-quality misaligned samples (Chen et al., 2025; Liu et al., 2025c), but only provide binary alignment labels without the granular diagnostic information required for training error categorization, localization, and correction models, and their limited scale proves insufficient. To address this gap, we synthesize misaligned pairs with complete diagnostic annotations based on the SCI Error Taxonomy, enabling systematic error injection across all 28 categories.

### 4.1. Task Definition

Let $\mathcal{S}$ denote an informal mathematical statement in natural language and $\mathcal{F}$ be its candidate formalization. We formulate the evaluation task as a conditional generation problem where FORMALRX maps the pair $(\mathcal{S}, \mathcal{F})$ to a structured diagnostic sequence $Y$. Each training instance $y_i$ and the output sequence $Y$ consists of four components (Table 2).

*Table 2.* Diagnostic components in FORMALRX.

| Component | Description |
| --- | --- |
| Verdict | Binary judgment (Aligned / Misaligned) |
| Categorization | Specific error type from SCI Error Taxonomy |
| Localization | Problematic code segment within $\mathcal{F}$ |
| Correction | Rectified code $\mathcal{F}'$ semantically aligned with $\mathcal{S}$ |

### 4.2. Data Source

We collect 17,825 aligned NL-FL pairs as seed data from two categories of sources as presented in Table 10. The first category comprises curated datasets and benchmarks from prior work, yielding 7,479 pairs. This includes small-scale datasets such as FIMO (Liu et al., 2023) and Comp-files (Renshaw & contributors, 2024), as well as established benchmarks spanning PutnamBench (Tsoukalas et al., 2024), ProverBench (Ren et al., 2025), CombiBench (Liu et al., 2025a), Gaokao-Formal (Yu et al., 2025a), and Formal-MATH (Yu et al., 2025b), which provide diverse mathematical domains with verified quality. To enhance diversity and coverage of real formalization scenarios, we incorporate a library category by identifying entries where natural language documentation precedes formal statements from Mathlib (The Mathlib Community, 2020). We retain only theorem statements that require proofs, excluding auxiliary definitions and helper constructs, finally yielding 10,346 supplementary seed pairs.

We deliberately exclude MiniF2F (Zheng et al., 2022) and ProofNet (Azerbayev et al., 2023) despite their widespread use in this field. Prior studies (Ospanov et al., 2025; Chen et al., 2025) have documented significant alignment issues in these benchmarks, with human experts identifying and annotating misaligned instances to construct Consistency-Check (Chen et al., 2025). We reserve ConsistencyCheck as an out-of-domain test set to evaluate generalization to real-world alignment errors. Using these benchmarks as seed data would introduce noise into our synthesis process, as errors in seed statements could propagate unpredictably during synthetic error generation. All collected seed data are unified to Lean v4.24.0 and verified using the Lean REPL (Leanprover Community, 2024). Compilation details are provided in Appendix E.2.

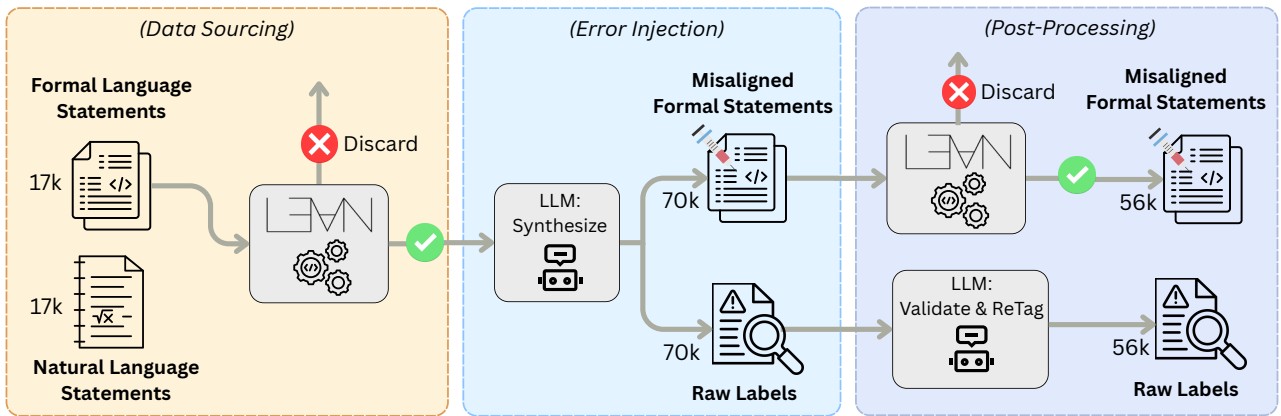

*Figure 3.* Overview of the data synthesis pipeline. From 17k aligned NL-FL seed pairs, an LLM applies taxonomy-guided error injection to produce misaligned candidates, and a second LLM validates and re-tags each against the SCI Taxonomy. Candidates failing Lean compilation or the label check are discarded, leaving 56k annotated samples. See Section 4.3 for details.

## 4.3. Data Synthesis

### 4.3.1. ERROR INJECTION

We employ taxonomy-guided error injection where an LLM agent systematically generates misaligned candidates from aligned seed pairs. Given an aligned NL-FL pair, the agent analyzes the formal statement against the SCI Error Taxonomy to identify applicable error categories, synthesizes a misaligned variant for each category by introducing the error type, and generates diagnostic metadata including error categorization, error localization, and explanatory content. For categories that could apply, priority ordering (Section 3.1) ensures deterministic assignment. Compared to rule-based perturbation (Lu et al., 2025) and manual annotation (Liu et al., 2025c) in prior work, taxonomy-guided injection captures subtle semantic distinctions requiring mathematical context understanding. Complete agent prompts are provided in Appendix I.4. Synthetic examples across categories are shown in Appendix G.2.

### 4.3.2. POST-PROCESSING

Generated instances are filtered through two-stage post-processing for code validity and label accuracy, retaining only samples that pass both. For code validity, all synthesized formal statements are verified through the Lean REPL (Leanprover Community, 2024), with compilation failures immediately discarded to isolate semantic misalignment from syntactic errors. For label accuracy, compiled instances undergo independent re-classification via an LLM-based agent (prompt in Appendix I.5). The agent analyzes the difference between modified and original statements, reassigning each instance to SCI categories with complete diagnostic annotations (error category, localization, reasoning, correction); examples are in Appendix G.3. Compilation statistics and re-classification details are provided in Appendix E.2 and C.2.

## 5. Setup

### 5.1. Training Setup

**Training Objective.** We employ standard Supervised Fine-Tuning using the negative log-likelihood loss over the target diagnostic sequence:

$$\mathcal{L}_{\text{SFT}}(\theta) = -\sum_{i=1}^{N}\sum_{t=1}^{T_i} \log p_\theta(y_{i,t} \mid \mathcal{S}_i, \mathcal{F}_i, y_{i,<t}) \quad (1)$$

where $y_{i,t}$ is the $t$-th token of the diagnostic sequence for the $i$-th training instance. The complete training prompt template is provided in Appendix I.1.

**Unified Generative Approach.** Unlike prior pipeline approaches (Wang et al., 2025a; Yu et al., 2025a), and method with multitask training loss (Lu et al., 2025), FORMALRX performs all diagnostic sub-tasks in a single forward pass. This unified architecture mitigates error propagation between cascaded stages and encourages the model to learn a shared representation that simultaneously captures global semantic characteristics and local syntactic correctness.

**Backbone Models.** We conducted experiments on three model sizes: Qwen3-4B-Instruct as the standard baseline, Qwen3-1.7B to validate small model capability on complex diagnostic tasks, Qwen3-8B as the primary model balancing performance and efficiency.

### 5.2. Evaluation Setup

**Evaluation Metrics.** We evaluate each task independently with appropriate metrics. Task 1 (Verdict) and Task 2 (Categorization) use classification metrics: accuracy, precision, recall, and macro F1-score for binary alignment and 28-class error type categorization respectively. Task 3 (Location) and Task 4 (Correction) employ a two-stage evaluation protocol (detailed below), reporting exact match counts, LLM-judged

*Table 3.* Main results on the in-domain test set. **Metrics (Task 1 & 2):** Acc = Accuracy, P = Precision, R = Recall, F1 = Macro F1-score. **Evaluation (Task 3 & 4):** ExMatch reports exact string match count; LLM indicates samples evaluated by LLM-as-judge; Acc represents final accuracy combining both exact match and LLM evaluation. All scores except ExMatch and LLM are reported on a 0-1 scale. Specialized metrics can only perform Task 1 (binary alignment).

| Model | Verdict | | | | Categorization | | | | Localization | | | Correction | | |
|---|---|---|---|---|---|---|---|---|---|---|---|---|---|---|
| | Acc | P | R | F1 | Acc | P | R | F1 | ExMatch | LLM | Acc | ExMatch | LLM | Acc |
| *Frontier Models* | | | | | | | | | | | | | | |
| DeepSeek-v3.2 | 0.754 | 0.752 | 0.998 | 0.858 | 0.157 | 0.376 | 0.198 | 0.259 | 1770 | 1094 | 0.407 | 2406 | 1752 | 0.592 |
| GPT-4.1 | 0.741 | 0.899 | 0.742 | 0.810 | 0.315 | 0.656 | 0.169 | 0.269 | 3261 | 963 | 0.609 | 2797 | 1275 | 0.579 |
| GPT-5-mini | 0.806 | 0.832 | 0.926 | 0.877 | 0.335 | 0.612 | 0.294 | 0.397 | 3164 | 1175 | 0.617 | 2299 | 1955 | 0.605 |
| GPT-5.2 | 0.779 | 0.821 | 0.899 | 0.858 | 0.273 | 0.529 | 0.221 | 0.311 | 2594 | 1378 | 0.565 | 977 | 3084 | 0.578 |
| GPT-5.3-codex | 0.791 | 0.897 | 0.812 | 0.853 | 0.376 | 0.734 | 0.256 | 0.380 | 2992 | 1680 | 0.654 | 2822 | 2013 | 0.688 |
| Claude-Sonnet-4 | 0.801 | 0.852 | 0.887 | 0.869 | 0.338 | 0.633 | 0.267 | 0.376 | 3272 | 822 | 0.582 | 3222 | 962 | 0.595 |
| Claude-Sonnet-4.6 | 0.828 | 0.917 | 0.845 | 0.880 | 0.448 | 0.814 | 0.335 | 0.475 | 2221 | 2924 | 0.739 | 4146 | 867 | 0.714 |
| *Instruct Models* | | | | | | | | | | | | | | |
| Qwen3-8B | 0.739 | 0.797 | 0.873 | 0.833 | 0.165 | 0.316 | 0.103 | 0.155 | 1815 | 984 | 0.398 | 2200 | 1002 | 0.458 |
| Qwen3-4B | 0.748 | 0.752 | 0.989 | 0.854 | 0.074 | 0.204 | 0.084 | 0.119 | 1054 | 1276 | 0.331 | 547 | 1941 | 0.354 |
| Qwen3-1.7B | 0.735 | 0.755 | 0.953 | 0.843 | 0.071 | 0.170 | 0.063 | 0.092 | 874 | 1584 | 0.350 | 1495 | 833 | 0.331 |
| *Specialized Alignment Metrics* | | | | | | | | | | | | | | |
| LeanScorer | 0.666 | 0.640 | 0.682 | 0.632 | – | – | – | – | – | – | – | – | – | – |
| BLEU (best F1) | 0.269 | 0.255 | 0.970 | 0.404 | – | – | – | – | – | – | – | – | – | – |
| FORMALRX-8B | 0.832 | 0.932 | 0.836 | **0.881** | 0.644 | 0.905 | 0.583 | **0.709** | 4563 | 707 | **0.750** | 4551 | 571 | **0.729** |

sample counts, and final accuracy.

**Two-Stage Evaluation Protocol.** For Tasks 3 and 4, we evaluate whether model predictions match the ground truth through a two-stage process. We first normalize whitespace and perform exact string matching, marking exact matches as correct. For non-matching pairs, we employ DeepSeek-v3.2(DeepSeek-AI et al., 2025b) to assess whether the prediction and ground truth are semantically equivalent. The process is necessary since while our model generates diagnostic outputs, the evaluation compares these outputs against reference annotations. Two location descriptions may refer to the same code segment with different wording, and two Lean correction statements may be logically equivalent despite syntactic differences. The LLM judge determines equivalence by comparing both outputs against the shared context of the informal problem and formal statement. We validate the reliability of this LLM-as-judge approach through human expert annotation in Appendix C, and evaluation prompt is provided in Appendix I.1.

**Baselines.** We compare our model against three categories of methods using a unified system prompt and structured output format. To ensure reliable parsing, all models operate in standard output mode under zero-shot settings.

- **Instruct Models:** To isolate the impact of our fine-tuning, we evaluate the Qwen3 backbone series (8B, 4B, 1.7B) (Yang et al., 2025).
- **Frontier Models:** state-of-the-art LLMs spanning DeepSeek-V3.2 (DeepSeek-AI et al., 2025b), GPT-

4.1 (OpenAI, 2025), GPT-5-mini (OpenAI, 2025a), GPT-5.2 (OpenAI, 2025b), GPT-5.3-codex (OpenAI, 2026), Claude-Sonnet-4 (Anthropic, 2025), and Claude-Sonnet-4.6 (Anthropic, 2026).
- **Alignment Metrics:** LeanScorer (Yu et al., 2025a) decomposes statements into premises and conclusions for fine-grained scoring, while BLEU (Papineni et al., 2002) measures surface-level token similarity. These methods provide only binary alignment verdicts.

## 6. Experimental Results

### 6.1. Main Results

Table 3 reports a comprehensive comparison against baselines across all four diagnostic tasks, with underlined entries marking the best within each category and bold entries the overall best. FORMALRX-8B attains the highest score on every task.

**Task 1: Verdict Prediction.** Among baselines, Claude-Sonnet-4.6 attains the strongest F1 of 0.880, ahead of Claude-Sonnet-4 (0.869) and DeepSeek-v3.2 (0.858). FORMALRX-8B reaches 0.881, marginally exceeding the strongest baseline and setting a new state of the art. The specialized metrics LeanScorer (0.632) and BLEU (0.404) lag substantially, underscoring the inadequacy of heuristic alignment signals.

**Task 2: Error Categorization.** Fine-grained classification over 28 categories is markedly harder, with no baseline

*Table 4.* Performance comparison across different model sizes. **Metrics:** Acc = Accuracy, P = Precision, R = Recall, F1 = F1 score. **Evaluation:** ExMatch reports exact string match count; LLM indicates samples evaluated by LLM-as-judge; Acc represents final accuracy combining both exact match and LLM evaluation. All scores except ExMatch and LLM are reported on a 0-1 scale.

| Model | Verdict | | | | Categorization | | | | Location | | | Correction | | |
|---|---|---|---|---|---|---|---|---|---|---|---|---|---|---|
| | Acc | P | R | F1 | Acc | P | R | F1 | ExMatch | LLM | Acc | ExMatch | LLM | Acc |
| FORMALRX-8B | 0.832 | 0.932 | 0.836 | 0.881 | 0.644 | 0.905 | 0.583 | 0.709 | 4563 | 707 | 0.750 | 4551 | 571 | 0.729 |
| FORMALRX-4B | 0.837 | 0.901 | 0.878 | 0.889 | 0.476 | 0.804 | 0.393 | 0.508 | 4478 | 534 | 0.713 | 3906 | 689 | 0.654 |
| FORMALRX-1.7B | 0.830 | 0.919 | 0.847 | 0.882 | 0.653 | 0.890 | 0.611 | 0.725 | 4474 | 721 | 0.739 | 4470 | 504 | 0.708 |

exceeding 0.476 in F1. Claude-Sonnet-4.6 leads among baselines (0.475), followed by GPT-5.3-codex (0.380) and Claude-Sonnet-4 (0.376). FORMALRX-8B attains 0.709, a 23.4-point improvement over the strongest baseline. Base models without diagnostic supervision remain near random (0.092–0.155), indicating that taxonomy-grounded categorization is not learnable from general instruction tuning alone.

**Task 3: Error Localization.** Claude-Sonnet-4.6 is the strongest baseline at 0.739, while the best instruction-tuned model (Qwen3-8B) reaches only 0.398. FORMALRX-8B attains 0.750, surpassing the strongest commercial baseline by 1.1 points and the strongest base model by 35.2 points. The narrow margin over Claude-Sonnet-4.6 suggests that recent frontier models have begun to acquire localization capability through scale, though specialized training remains advantageous.

**Task 4: Correction.** Claude-Sonnet-4.6 leads the baselines at 0.714, while Qwen3-8B reaches only 0.458. FORMALRX-8B attains 0.729, exceeding the strongest commercial baseline by 1.5 points and the strongest base model by 27.1 points. The wide gap between instruction-tuned and frontier models indicates that generating syntactically valid Lean statements that preserve the intended semantics demands targeted supervision rather than emerging naturally from scale.

FORMALRX-8B consistently achieves the highest performance across all four diagnostic tasks. Performance gaps vary by difficulty: modest improvements on Task 1 (Verdict) where base models already perform well (0.833–0.854), and a substantial advantage on Task 2 (Categorization) where FORMALRX-8B outperforms the strongest baseline by 23.4 percentage points. While recent frontier models such as Claude-Sonnet-4.6 have narrowed the gap on Tasks 3 and 4, FORMALRX-8B remains the best overall as a compact 8B model fine-tuned on diagnostic data. Specialized metrics like LeanScorer and BLEU are limited to binary classification and cannot perform diagnostic tasks beyond Verdict. These results validate that while modern LLMs possess inherent alignment detection capabilities, comprehensive diagnostic reasoning necessitates task-specific supervised training.

### 6.2. Out-of-Domain Generalization

We further evaluate FORMALRX on an out-of-domain benchmark ConsistencyCheck (Chen et al., 2025), which contains naturally misaligned statements beyond the training distribution. As shown in Table 5, FORMALRX-8B achieves an F1 of 0.596, the strongest result among instruction-tuned base models and outperforming LeanScorer (0.588), though falling short of frontier models. The drop relative to in-domain performance is consistent with two compounding distribution shifts (Moreno-Torres et al., 2012): ConsistencyCheck has a positive-to-negative ratio of roughly 2:1, whereas our training data follows an approximately 1:3 ratio, and its misalignments arise naturally rather than through systematic synthesis, which may yield different linguistic characteristics from our training errors.

*Table 5.* Out-of-domain evaluation on ConsistencyCheck.

| Method | Accuracy | Precision | Recall | F1 |
|---|---|---|---|---|
| FORMALRX-8B | 0.702 | 0.548 | 0.654 | 0.596 |
| FORMALRX-4B | 0.672 | 0.509 | 0.689 | 0.585 |
| FORMALRX-1.7B | 0.653 | 0.490 | 0.730 | 0.586 |
| Qwen3-8B | 0.668 | 0.506 | 0.640 | 0.565 |
| Qwen3-4B | 0.583 | 0.430 | 0.734 | 0.542 |
| Qwen3-1.7B | 0.536 | 0.393 | 0.699 | 0.503 |
| DeepSeek-v3.2 | 0.858 | 0.652 | 0.698 | 0.674 |
| GPT-4.1 | 0.774 | 0.725 | 0.529 | 0.612 |
| GPT-5-mini | 0.840 | 0.816 | 0.677 | **0.740** |
| GPT-5.2 | 0.710 | 0.551 | 0.872 | 0.675 |
| GPT-5.3-Codex | 0.822 | 0.726 | 0.726 | 0.726 |
| Claude-Sonnet-4.6 | 0.829 | 0.924 | 0.490 | 0.640 |
| LeanScorer | 0.768 | 0.732 | 0.491 | 0.588 |
| BLEU (Best F1) | 0.413 | 0.359 | 0.945 | 0.520 |

The ranking of frontier models on this benchmark also warrants careful interpretation. GPT-5-mini achieves the highest F1 of 0.740, yet on the in-domain test set it substantially underperforms FORMALRX on Tasks 2 through 4. This inconsistency suggests that performance on ConsistencyCheck partly reflects sensitivity to its specific label distribution rather than general diagnostic capability. We view out-of-domain evaluation on naturally annotated data as a valuable but inherently noisy signal, and leave improving cross-distribution robustness to future work.

## 6.3. Scaling Analysis

Table 4 reports FORMALRX across three model sizes. FOR-MALRX-8B achieves the strongest overall performance, with F1 of 0.881 (Verdict) and 0.709 (Categorization), and accuracy of 0.750 (Localization) and 0.729 (Correction).

Scaling behavior differs across tasks. On Verdict, all three sizes exceed 0.88 in F1, with FORMALRX-4B marginally ahead (0.889 vs. 0.881), indicating that binary alignment is well-learned even at 1.7B parameters. On Categorization, FORMALRX-1.7B leads at 0.725 while FORMALRX-4B drops to 0.508, suggesting that 28-class categorization is sensitive to factors beyond raw scale; FORMALRX-8B at 0.709 remains within 1.6 points of the best.

The generative tasks show clearer scaling. FORMALRX-8B attains 0.750 on Localization (vs. 0.713 and 0.739 for 4B and 1.7B) and 0.729 on Correction (vs. 0.654 and 0.708), confirming that producing syntactically valid Lean code with correct semantics benefits from increased capacity. For Localization, 4563 samples (64.9%) pass exact match and 707 (10.1%) are recovered by LLM judging, illustrating that the two-stage protocol is essential since exact matching alone would miss roughly 10% of semantically correct outputs.

## 7. Conclusion

We presented FORMALRX, an evaluation framework that transforms autoformalization assessment from binary judgments into actionable diagnostics. By introducing the SCI, a hierarchical classification of 28 error categories, we enable the systematic decomposition of formalization failures that previously went undiagnosed. Unlike existing paradigms, FORMALRX provides precise error localization and corrective guidance, bridging the gap between evaluation and model refinement. These high-fidelity signals offer a path toward more nuanced RLHF pipelines and iterative system improvements. Ultimately, FORMALRX establishes a transparent foundation for moving autoformalization beyond black-box assessments toward instructive and verifiable quality assurance.

## Impact Statement

This paper introduces FORMALRX, a framework designed to improve the reliability of autoformalization by providing interpretable, fine-grained diagnostics for mathematical reasoning. By transitioning from opaque binary evaluations to the structured SCI Error Taxonomy, our work enhances the transparency of AI-driven formal verification, directly addressing the critical issues of hallucinations and unfaithful reasoning in LLMs. The primary societal impact lies in strengthening the bridge between intuitive natural language

and rigorous formal logic, which is essential for deploying AI in safety-critical domains such as hardware verification, software synthesis, and automated theorem proving. While more effective error localization could theoretically be repurposed to find vulnerabilities in formal specifications, the overarching goal is to provide dense supervision signals that foster the development of more robust, verifiable, and trustworthy AI systems. There are many potential societal consequences of our work, none of which we feel must be specifically highlighted beyond these advancements in model interpretability and logical alignment.

## Limitations

Despite its contributions to diagnostic evaluation in autoformalization, FormalRX has several limitations that warrant careful consideration.

**Manual Curation of Taxonomy.** The SCI error taxonomy is constructed through manual expert inspection and iterative refinement over a limited set of examples, which introduces potential subjectivity and limits scalability. Although designed to be comprehensive and disjoint as well as having its effectiveness verified by human annotation results and training experiments, it is difficult to guarantee coverage of long-tail or emergent failure modes in real-world settings. This also constrains full transfer of the existing framework to other proof assistant languages (e.g., Coq, Isabelle). Although the high-level taxonomy categories may generalize, their instantiation and prioritization may not fully resemble the observed patterns in Lean autoformalization systems.

**Limited robustness evaluation.** Empirical evaluation is primarily conducted on in-domain or closely related datasets, with only limited evaluation on the out-of-distribution (OOD) benchmarks. On the one included OOD benchmark, the performance advantage of FormalRX narrows, raising concerns about generalization in the realistic settings. Broader evaluation across diverse formalization corpora, domains, and error distributions can be included to establish robustness.

Addressing these limitations, particularly improving real-world data coverage and strengthening robustness evaluation, constitutes an important direction for future work.

## Acknowledgement

This work is partially supported by the Swiss AI Initiative Small Compute Grant. Yinya Huang is supported by an ETH AI Center Postdoctoral Fellowship.

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

## A. Implementation Details

### A.1. Training Configuration

Table 6 summarizes the key hyperparameters used for training FORMALRX-8B. The model is trained using the AdamW optimizer (Loshchilov & Hutter, 2019) with a cosine learning rate scheduler and DeepSpeed ZeRO-2 (Rajbhandari et al., 2020) for efficient distributed training.

*Table 6.* Training hyperparameters

| Hyperparameter | Value |
|---|---|
| Finetuning Type | Full |
| Optimizer | AdamW |
| Learning Rate | 2.0e-5 |
| LR Scheduler | Cosine |
| Warmup Ratio | 0.05 |
| Training Epochs | 10 |
| Per-device Batch Size | 8 |
| Gradient Accumulation Steps | 2 |
| Effective Batch Size | 1,024 (64 GPUs) |
| Max Sequence Length | 3,072 |
| Precision | BF16 |
| DeepSpeed Stage | ZeRO-2 |

### A.2. Hardware and Training Time

All experiments are conducted on 16 nodes of NVIDIA GH200 GPUs. Training FORMALRX-8B for 10 epochs on 56,287 samples takes approximately 2 hours. Model checkpoints are saved every 100 steps with the best model selected based on validation error categorization F1-score.

### A.3. Input Format

We adopt the ShareGPT conversation format to handle the extensive context required by the SCI Error Taxonomy. The unified input sequence integrates the taxonomy definition $\mathcal{T}$, informal statement $\mathcal{S}$, and formal code $\mathcal{F}$, with a maximum context length of 3,072 tokens. This format preserves the full 28-category taxonomy in the system prompt without truncation and enhances instruction-following for multi-part diagnostic outputs.

## B. Ablation Study

### B.1. Base Model Selection

To investigate how different base model architectures respond to our FORMALRX framework, we fine-tune three models using identical training configurations: Qwen2.5-7B (Qwen et al., 2025), DeepSeek-R1-Distill-Qwen-7B (Guo et al., 2025a), and Qwen3-8B (Yang et al., 2025). As shown in Table 7, different base models exhibit distinct learning patterns across the diagnostic tasks.

*Table 7.* Performance comparison of FORMALRX fine-tuned on different base models on FORMALRX-test (7030 samples).

| Base Model | Verdict | | | | Categorization | | | | Localization | | | Correction | | |
|---|---|---|---|---|---|---|---|---|---|---|---|---|---|---|
| | Acc | P | R | F1 | Acc | P | R | F1 | ExMatch | LLM | Acc | ExMatch | LLM | Acc |
| Qwen2.5-7B | 0.840 | 0.922 | 0.857 | 0.889 | 0.671 | 0.897 | 0.631 | 0.741 | 4672 | 678 | 0.761 | 3575 | 529 | 0.584 |
| DeepSeek-R1-Distill-7B | 0.797 | 0.924 | 0.793 | 0.854 | 0.625 | 0.896 | 0.562 | 0.691 | 4267 | 771 | 0.716 | 4853 | 498 | 0.761 |
| Qwen3-8B | 0.832 | 0.932 | 0.836 | 0.881 | 0.644 | 0.905 | 0.583 | 0.709 | 4563 | 707 | 0.750 | 4551 | 571 | 0.729 |

Best results are shown in underline.

Qwen2.5-7B demonstrates strong adaptation to classification-based diagnostic tasks (Tasks 1-3), achieving the highest F1-scores of 0.889 and 0.741 for verdict and categorization, and 0.761 accuracy for localization. However, it shows limited improvement on correction generation (Task4), reaching only 0.584 accuracy. This suggests that while its capabilities suffice for diagnostic tasks, the generation capacity may be insufficient for producing correct formal code even after fine-tuning.

DeepSeek-R1-Distill-Qwen-7B achieves the best correction performance (0.761), likely benefiting from its reasoning-focused pre-training, but shows weaker diagnostic task performance. Qwen3-8B strikes the best balance, maintaining competitive diagnostic performance while significantly outperforming Qwen2.5-7B on correction (0.729 vs 0.584, +24.8%).

We adopt Qwen3-8B as our primary model for the following considerations: (1) **Output reliability**: DeepSeek-R1-Distill-Qwen-7B, designed as a reasoning-focused model with chain-of-thought mechanisms, occasionally deviates from the structured output format required by our diagnostic framework, producing unparseable responses that disrupt the automated pipeline—in contrast, Qwen3-8B demonstrates consistent adherence to the specified system prompt format; (2) **Balanced diagnostic performance**: while DeepSeek-R1 achieves the highest correction accuracy of 0.761, it shows degradation in earlier stages with F1-score of 0.854 for verdict compared to Qwen3-8B's 0.881, and 0.691 for categorization compared to 0.709; (3) **Generation capability**: Qwen3-8B substantially outperforms Qwen2.5-7B on correction, achieving 0.729 accuracy with a 24.8% improvement over the 0.584 baseline.

### B.2. Training Method Selection

Fixing Qwen3-8B (Yang et al., 2025) as the base model and the training data of Section B.1, we compare progressive training against three alternatives: joint training, which optimizes all four tasks at once; single-task training, which trains one independent model per task; and sequential training, which introduces tasks one at a time without data replay. Results are reported in Table 8.

*Table 8.* Training method comparison on FORMALRX-Test (7,030 samples) with Qwen3-8B. Single-Task uses four separate models and is a per-task upper bound; Sequential numbers are measured on each target task in isolation, as unified output is lost to catastrophic forgetting. ExMatch and LLM are sample counts; Acc, P, R, F1 are rates. Bold marks the best unified single-model result.

| Method | Verdict (T1) | | | | Categorization (T2) | | | | Localization (T3) | | | Correction (T4) | | |
|---|---|---|---|---|---|---|---|---|---|---|---|---|---|---|
| | Acc | P | R | F1 | Acc | P | R | F1 | ExMatch | LLM | Acc | ExMatch | LLM | Acc |
| *Unified single model* | | | | | | | | | | | | | | |
| Joint Training | 0.832 | 0.932 | 0.836 | 0.881 | 0.644 | 0.905 | 0.583 | 0.709 | 4563 | 707 | 0.750 | 4551 | 571 | 0.729 |
| Progressive Training | **0.853** | 0.923 | 0.877 | **0.899** | **0.692** | 0.900 | 0.859 | **0.761** | 4741 | 731 | **0.778** | 5107 | 460 | **0.792** |
| *Reference (isolated, not unified)* | | | | | | | | | | | | | | |
| Single-Task[†] | 0.814 | 0.860 | 0.897 | 0.878 | 0.696 | 0.930 | 0.641 | 0.759 | 4839 | 908 | 0.818 | 5646 | 416 | 0.862 |
| Sequential[‡] | 0.814 | 0.860 | 0.897 | 0.878 | 0.675 | 0.952 | 0.595 | 0.732 | 4731 | 551 | 0.751 | 5446 | 403 | 0.832 |

[†] Four independent models, one per task.  [‡] Each row measured on its target task only.

**Progressive vs. joint training.** Both produce all four diagnostic fields from a single model, so this contrast isolates the effect of introducing tasks incrementally with replay rather than optimizing every objective from the outset. Progressive training wins on all four tasks, with the largest gains on correction (0.729 to 0.792) and categorization F1 (0.709 to 0.761), and consistent improvements on verdict F1 (0.881 to 0.899) and localization (0.750 to 0.778).

**Progressive vs. single-task training.** Single-task training dedicates one model to each task and serves as a per-task upper bound, at the cost of four separate models and no unified inference. Progressive training matches or exceeds it on verdict (0.899 vs. 0.878) and categorization F1 (0.761 vs. 0.759), and trails only on localization (0.778 vs. 0.818) and correction (0.792 vs. 0.862). A single unified model thus recovers most of the specialized upper bound while retaining joint diagnostic inference.

**Progressive vs. sequential training.** Sequential training fine-tunes on each newly introduced task without revisiting prior data. Measured on each target task in isolation it stays competitive and even leads on correction (0.832), but it suffers catastrophic forgetting: later stages overwrite the structured output format, so the model loses the ability to emit all four fields jointly. Example below traces this degradation stage by stage. By replaying earlier tasks, progressive training preserves full diagnostic capability while reaching 0.899 verdict F1.

---

**Example: Catastrophic Forgetting in Sequential Training**

**Informal Problem.** If $f$ is any function and $g$ is analytic without zero in $U$, then $f$ is meromorphic in normal form on $U$ iff $f \cdot g$ is meromorphic in normal form on $U$.

**Formal Statement.**

```
theorem meromorphicNFOn_mul_iff_left_of_analyticOnNhd {f g : K → K}
    (h₁g : AnalyticOnNhd K g U)
    (h₂g : ∀ u ∈ U, g u ≠ 0) :
    MeromorphicNFOn (f + g) U ↔ MeromorphicNFOn f U := sorry
```

**Expected Output.**

```
Semantic Alignment: Misaligned
Error Type: Operator Confusion
Error Location: MeromorphicNFOn (f + g) U
Corrected Statement:
theorem meromorphicNFOn_mul_iff_left_of_analyticOnNhd {f g : K → K}
    (h₁g : AnalyticOnNhd K g U)
    (h₂g : ∀ u ∈ U, g u ≠ 0) :
    MeromorphicNFOn (f ⋆ g) U ↔ MeromorphicNFOn f U := sorry
```

**After T2** (verdict field lost).

```
Error Type: Operator Confusion
```

**After T3** (error type field corrupted by location content).

```
Error Type: MeromorphicNFOn (f + g) U
```

**After T4** (error type field overwritten by corrected statement).

```
Error Type:
theorem meromorphicNFOn_mul_iff_left_of_analyticOnNhd {f g : K → K}
    (h₁g : AnalyticOnNhd K g U) (h₂g : ∀ u ∈ U, g u ≠ 0) :
    MeromorphicNFOn (f ⋆ g) U ↔ MeromorphicNFOn f U := sorry
```

---

# C. Human Study

## C.1. LLM-Based Components Validation

To validate the reliability of LLM-based components in our diagnostic pipeline, we conduct expert evaluation across all four diagnostic stages: (1) negative example synthesis, (2) output validation and re-tag, (3) error localization judgment, and (4) correction judgment.

### C.1.1. EXPERIMENTAL SETUP

**Task Design** We evaluate four critical LLM-based component in our pipeline:

- **Task 1 (Synthesis):** Given a correct formal statement, can the LLM select appropriate error types from the SCI Error Taxonomy and generate modified statements that successfully introduce the target error?
- **Task 2 (Validation and Re-tag):** Given a statement pair and an initial error categorization, can the LLM assess the accuracy of the initial error categorization and produce a validated SCI Error Taxonomy tag with appropriate error type, localization, reasoning, and correction suggestions?
- **Task 3 (Localization Judge):** Can the LLM correctly determine whether two error location descriptions refer to the same buggy code region?
- **Task 4 (Correction Judge):** Can the LLM correctly determine whether two formal statements are semantically equivalent?

*Table 9.* Human study results validating LLM components in the diagnostic pipeline. Accuracy computed on all annotated samples (n=240 per task); agreement and $\kappa$ on the double-annotated subset (n=38 per task).

| Task | Accuracy | Agreement | Disagreements | $\kappa$ (avg) |
|---|---|---|---|---|
| T1: Synthesis | 78.3% (188/240) | 84.2% (32/38) | 6 | 0.684 |
| T2: Validation & Re-tag | 86.2% (207/240) | 94.7% (36/38) | 2 | 0.895 |
| T3: Localization Judge | 84.6% (203/240) | 86.8% (33/38) | 5 | 0.789 |
| T4: Correction Judge | 89.6% (215/240) | 92.1% (35/38) | 3 | 0.842 |
| **Overall** | **84.7% (813/960)** | **89.5% (136/152)** | **16** | **0.803** |

**Data Collection** We sample instances per task from our synthetic dataset pipeline, stratified across error categories and difficulty levels. Each instance contains the output from LLM alongside ground truth, with experts evaluating whether the judgment or generation from LLM is correct.

**Annotation Protocol** We recruited additional Lean experts to scale the validation. Each task is annotated with 240 expert judgments, of which 38 come from double-annotated samples (each labeled by two experts under a rotating overlap design) used to measure inter-annotator agreement (Krippendorff, 2018). This yields 960 judgments and 152 double-annotated pairs across the four tasks. Annotators (backgrounds: automated theorem proving, natural language processing, mathematics) receive the full SCI Error Taxonomy (Appendix F.2), task-specific guidelines, and calibration on pilot examples. For each sample, annotators provide binary judgments (correct=1, incorrect=0) with optional explanatory notes.

C.1.2. RESULTS

Table 9 presents results across all tasks. LLM accuracy ranges from 78.3% (Task 1: Synthesis) to 89.6% (Task 4: Correction Judge), with an overall accuracy of 84.7% (813/960), demonstrating strong performance in automated diagnostics. Inter-annotator agreement on double-annotated samples ranges from 84.2% to 94.7%, with an overall agreement of 89.5% (136/152), indicating substantial consistency in expert judgments.

**Reliability Analysis** We report both percentage agreement and Cohen's $\kappa$ as inter-annotator agreement metrics. With the expanded annotation set, per-task $\kappa$ ranges from 0.684 to 0.895 (overall 0.803), corresponding to substantial-to-almost-perfect agreement. Task 2 attains the highest $\kappa$ (0.895), consistent with its high percentage agreement, while Task 1 shows the lowest (0.684), reflecting the greater subjectivity of judging synthesized error injection. The agreement between the percentage and $\kappa$ measures, now stable under the larger sample, supports the reliability of our annotation setup.

**Error Analysis** Analyzing the 16 disagreement cases across the double-annotated samples, we find that most involve borderline judgments where LLM outputs are technically correct but suboptimal. For example, in Task 3, some error location descriptions were overly verbose yet semantically accurate; in Task 2, certain error categorizations were conservative but valid. A smaller number reflect genuine annotation differences due to guideline ambiguity, which informed refinements to our task instructions. The low disagreement rate and predominance of edge cases indicate that our diagnostic criteria are well-defined and consistently interpretable by experts.

**Per-Task Analysis**

The performance gap between Task 1 (78.3%) and Task 2 (86.2%) demonstrates the critical role of Task 2 as a validation and correction mechanism. Task 2 (Validation and Re-tag) achieves the highest inter-annotator agreement (94.7%) and $\kappa$ (0.895), validating that our LLM can reliably apply the SCI Error Taxonomy to verify and refine error categorizations from the synthesis stage. Tasks 3 and 4 (both LLM-as-judge) achieve 84.6% and 89.6% accuracy respectively, confirming the effectiveness of our judge prompts for evaluating localization and correction quality.

**C.2. Synthesis Quality Validation**

To verify the reliability of the synthetically generated data, we conducted a limited manual inspection of 200 randomly sampled instances drawn proportionally across all 28 error categories. Each instance included both the synthesized statement and its assigned SCI Error Taxonomy label.

**Validation Results.** Human evaluation indicates that 98% of sampled instances (196/200) are correctly labeled according to the SCI Error Taxonomy, with most inconsistencies occurring at boundary cases between nearby categories (e.g., S2.1

Object Type Errors vs. C1.3 Domain Constraint Errors). Although the validation covers only a small subset, the results suggest that the taxonomy-guided synthesis pipeline yields high-quality annotations with minimal noise, supporting the reliability of the training data described in Section 4.

## D. Baseline Reproduction

We reproduce two baseline methods for autoformalization evaluation, LeanScorer and BLEU-based alignment detection.

### D.1. LeanScorer

We reproduce LeanScorer (Yu et al., 2025a) as our primary baseline for evaluating autoformalization semantic correctness.

**Implementation.** Following the original paper, we implement the two-stage LeanScorer pipeline: (1) subtask decomposition via LLM prompt, and (2) subtask-level evaluation with three-tiered ratings (Perfectly Match, Minor Inconsistency, Major Inconsistency), aggregated through Sugeno Fuzzy Integral. Both stages employ DeepSeek-V3 (DeepSeek-AI et al., 2025a) with the prompt templates provided in Appendix B.5 of the original paper. The fuzzy measure is implemented according to Equation (3) in (Yu et al., 2025a), with decision threshold $\alpha = 0.6$ for binary classification.

**Consistency with original results.** Our reproduction achieves comparable performance to the reported results in the original paper, validating the reliability of our implementation.

### D.2. BLEU Metric

We implement BLEU (Papineni et al., 2002) as a surface-level baseline for binary alignment detection. For each natural language-formal statement pair, we compute sentence-level BLEU score and classify as aligned if BLEU exceeds a threshold. Since our dataset exhibits class imbalance (26.9% aligned), we perform threshold sweep over [0, 100] and select the threshold maximizing F1 score.

#### D.2.1. BLEU CALCULATION EXAMPLES

We present two representative cases illustrating why lexical overlap metrics fail to capture semantic alignment in autoformalization.

**Case 1: Aligned Pair (BLEU=0.74)**

*Informal Problem:* If we have a functor `F : J ==> Pi i, C i` into a category of indexed families, and we have limits for each of the `F >> Pi.eval C i`, then `F` has a limit.

*Formal Statement:*

```
theorem hasLimit_of_hasLimit_comp_eval : HasLimit F := sorry
```

*Analysis:* Despite being semantically aligned, this pair achieves only BLEU=0.74 due to severe vocabulary mismatch. The formal statement uses concise type-theoretic notation while the informal statement uses natural language. BLEU breakdown shows 1-gram precision of 25% but rapidly degrading n-gram precision (2-gram: 6.67%, 3-gram: 3.57%), with brevity penalty of 0.127 due to length disparity (49 vs 16 tokens). Token analysis reveals only 3.3% overlap, demonstrating that formal mathematics employs distinct vocabulary from informal descriptions even when semantically faithful.

**Case 2: Misaligned Pair (BLEU=1.15)**

*Informal Problem:* "If 'f' is any function and 'g' is analytic without zero in 'U', then 'f' is meromorphic in normal form on 'U' iff 'f * g' is meromorphic in normal form on 'U'."

*Formal Statement:*

```
theorem meromorphicNFOn_mul_iff_left_of_analyticOnNhd
  {f g : K -> K} (h1g : AnalyticOnNhd K g U)
  (h2g : forall u in U, g u != 0) :
  MeromorphicNFOn (f + g) U ←> MeromorphicNFOn f U := sorry
```

*Analysis:* This semantically misaligned pair (operator confusion: $f * g \rightarrow f + g$) scores BLEU=1.15, higher than the aligned case. The critical error—replacing multiplication with addition—is a single-token substitution that minimally affects lexical overlap. BLEU's n-gram matching (1-gram: 17.86%) captures superficial token similarity while completely missing the semantic error. Token analysis shows 0% overlap between informal and formal vocabularies, yet the score remains non-zero due to BLEU's smoothing mechanisms.

### D.2.2. THRESHOLD SELECTION STRATEGY

Figure 4 shows precision, recall, accuracy, and F1 across threshold values [0, 100]. The dataset's class imbalance (73.1% misaligned, 26.9% aligned) causes accuracy-optimized threshold (40.0) to achieve 74.5% accuracy but only F1=0.008, essentially predicting all negative. The F1-optimized threshold (0.0) achieves balanced F1=0.404 with precision=0.255 and recall=0.970. All methods substantially underperform learned approaches, confirming that surface-level lexical metrics are inadequate for autoformalization evaluation.

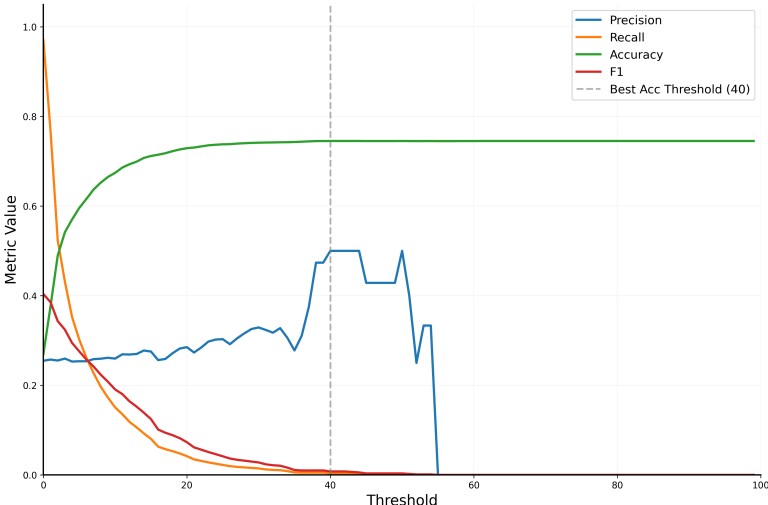

*Figure 4.* BLEU threshold sweep showing precision, recall, accuracy, and F1. Best accuracy threshold (40.0) yields degenerate classifier; best F1 threshold (0.0) provides meaningful baseline.

## E. Data Analysis

### E.1. Data Source

The seed data to synthesize negative examples is collected from existing NL–FL pair datasets, as presented in Table 10. Sources are categorized into three types: datasets, benchmarks and libraries.

*Table 10.* Data sources

| Category | Source | Count |
| --- | --- | --- |
| Dataset | FIMO | 149 |
| | Compfiles | 245 |
| Benchmark | PutnamBench | 659 |
| | ProverBench | 325 |
| | CombiBench | 101 |
| | Gaokao-Formal | 495 |
| | Formal-MATH | 5,505 |
| Library | Mathlib | 10,346 |
| **Total** | | **17,825** |

The dataset category consists of FIMO (Liu et al., 2023) and Compfiles (Renshaw & contributors, 2024). Since Compfiles is

under continuous development, we selected the commit dated November 2, 2025.

Benchmark data include PutnamBench (Tsoukalas et al., 2024), ProverBench (Ren et al., 2025), CombiBench (Liu et al., 2025a), Gaokao-Formal (Yu et al., 2025a), and FormalMATH (Yu et al., 2025b). For the library category, we extract NL–FL pairs from Mathlib (The Mathlib Community, 2020), retaining only the content strictly required by the target theorem and excluding auxiliary lemmas, unrelated definitions, and proof-specific constructs. In total, 10,346 NL–FL pairs are extracted and pass compilation checks. The collected data are semantically aligned and serve as ground-truth for synthesis.

We deliberately exclude MiniF2F (Zheng et al., 2022) and ProofNet (Azerbayev et al., 2023) despite their widespread use in training Lean models. Prior studies (Ospanov et al., 2025; Chen et al., 2025) have reported significant misalignment issues in these benchmarks, with human experts constructing ConsistencyCheck (Chen et al., 2025) from annotated misaligned instances as a high-quality binary classification dataset. Although Gaokao-Formal and FormalMATH are autoformalized, the authors claim manual verification ensures their quality.

### E.2. Compilation Detail

#### E.2.1. COMPILATION CHECK

We synthesize 70,832 negative examples (29,012 from benchmarks/datasets, 41,820 from Mathlib). All instances are verified through the Kimina Lean Server (Santos et al., 2025) under Lean version 4.24.0, retaining only successfully compiled samples. Of the 70,832 synthesized instances, 56,673 (80.0%) pass compilation, as shown in Table 11.

*Table 11.* Compilation results

| Source | Synthesized | Compiled | Success Rate |
|---|---|---|---|
| Benchmark/Dataset | 29,012 | 23,472 | 80.9% |
| Mathlib | 41,820 | 33,201 | 79.4% |
| **Total** | **70,832** | **56,673** | **80.0%** |

Two main categories of compilation failures occur. First, deprecated syntax in Lean 4.24.0 requires manual correction (e.g., replacing *in* with $\in$ in summation expressions like $\sum i$ *in Finset.Icc 1 10, ...*). Second, Mathlib-extracted data causes redefinition errors when verified through Kimina Lean Server due to dependency conflicts with the entire Mathlib repository, requiring alternative verification methods. Representative failure cases are analyzed in Appendix G.4.

#### E.2.2. COMPILATION FAILED CASES

**Case 1: FormalMATH *Complex.abs* deprecation**

A case from FormalMATH (Yu et al., 2025b). This statement failed to pass compilation check because *Complex.abs* is deprecated in latest version. It should be changed to $\|z\|$ instead. Most of compilation failures fall to such version-related category.

```
import Mathlib

open Complex Filter Function Metric Finset
open scoped BigOperators Topology

theorem olymid_ref_base_5957 :
  Set.ncard {z : ℂ | Complex.abs z = 1 ∧ 1 + z^5 + z^10 + z^15 + z^18 + z^21 +
    z^24 + z^27 = 0} = 11 := by sorry
```

**Case 2: PutnamBench *range* confusion**

Another case from PutnamBench (Tsoukalas et al., 2024). This case fails because of the confusion of *range*. There exists two function: *Set.range* and *Finset.range*. The first function denotes the image of a function as a set, while the second one denotes a computable finite set of natural numbers for enumeration.

```
import Mathlib

open Filter Topology Set

theorem putnam_2020_b1
(d : ℕ → ℕ)
(S : ℤ)
(hd : d = fun n : ℕ => Σ i : Fin (Nat.digits 2 n).length, (Nat.digits 2 n)[i]!)
-- Origin: S = Σ k : Icc 1 2020
(hS : S = Σ k : range 2021,
    ((-1 : ℤ)^(d k))*(k : ℤ)^3)
: S % 2020 = ((1990) : ℕ ) := sorry
```

### Case 3: Mathlib extracted statement, Kimina redefinition error

A case from Mathlib extracted statement. It failed to pass Kimina Lean Server ([Santos et al., 2025](#))'s check, because Kimina defaultly imports *Mathlib*, leading to a redefined error. Therefore Mathlib extracted data should directly use Lean REPL to check its syntax correctness.

```
import Mathlib

import Mathlib.Algebra.TrivSqZeroExt
@[expose] public section
variable {R A B : Type*}
abbrev DualNumber (R : Type*) : Type _ :=  TrivSqZeroExt R R
def DualNumber.eps [Zero R] [One R] : DualNumber R :=  TrivSqZeroExt.inr 1

@[inherit_doc]
scoped[DualNumber] notation "ε" => DualNumber.eps

@[inherit_doc]scoped[DualNumber] postfix:1024 "[ε]" => DualNumber

open DualNumber

namespace DualNumber

open TrivSqZeroExt

theorem commute_eps_left [Semiring R] (x : DualNumber R) : Commute ε x := sorry
```

## E.3. Negative Example Analysis

### E.3.1. ERROR TYPE DISTRIBUTION

The synthesis process achieves comprehensive coverage across all 28 categories while reflecting their natural occurrence patterns. Figure 5 and Table 12 present the resulting distribution.

The distribution demonstrates that all categories are successfully instantiated with no single category exceeding 14%. As catch-all categories, C3.1 (Missing Premise) and C4 (Conclusion Errors) naturally exhibit higher proportions at 13.79% and 9.06% respectively. Even the least frequent category (S3.4) contains 58 instances, confirming that the SCI Error Taxonomy captures the full spectrum of error types in real-world autoformalization scenarios.

## E.4. Training Data Construction

We construct our training dataset by combining all verified positive samples from benchmarks, datasets and Mathlib with synthetically generated negatives annotated using the SCI Error Taxonomy in Section 4.3. All samples are randomly shuffled and split with an 8:1:1 ratio for train/validation/test sets. Table 13 summarizes the final dataset composition.

*Table 12.* Top 10 error categories by frequency

| Error Category | Count | Proportion |
|---|---|---|
| Missing Premise | 7,243 | 13.79% |
| Conclusion Error | 4,759 | 9.06% |
| Quantifier Weakening | 3,839 | 7.31% |
| Positivity Constraint Error | 3,803 | 7.24% |
| Logical Connective Misuse | 3,763 | 7.17% |
| Quantifier Strengthening | 2,965 | 5.65% |
| Coefficient/Constant Error | 2,828 | 5.39% |
| Function Confusion | 2,504 | 4.77% |
| Incorrect Premise | 2,373 | 4.52% |
| Redundant Premise | 2,321 | 4.42% |
| Other 18 categories | 16,123 | 30.70% |
| **Total** | **52,521** | **100.00%** |

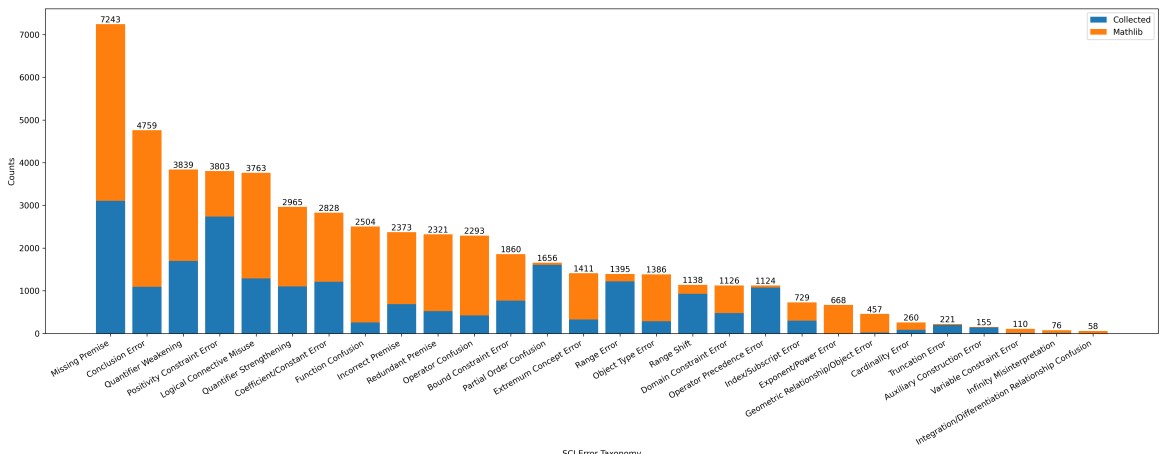

*Figure 5.* Distribution of synthesized instances across the 28 SCI Error Taxonomy categories.

*Table 13.* Dataset statistics. Of the 56,673 negatives that pass compilation (Table 11), 52,521 remain after re-tag filtering and enter the dataset.

| Source | Positive | Negative | Total |
|---|---|---|---|
| Benchmark/Dataset | 7,479 | 21,588 | 29,067 |
| Mathlib | 10,346 | 30,933 | 41,279 |
| **Total** | 17,825 | 52,521 | 70,346 |
| Train | 14,267 | 42,020 | 56,287 |
| Val | 1,779 | 5,250 | 7,029 |
| Test | 1,779 | 5,251 | 7,030 |

# F. SCI Error Taxonomy

## F.1. SCI Error Taxonomy Overview

*Table 14.* SCI Error Taxonomy Overview

| ID | Category | Description |
|---|---|---|
| **Semantic Error (S)** | | |
| S1.1 | Quantifier Strengthening | Quantifier modifications or domain expansions lead to over-strong claims. |
| S1.2 | Quantifier Weakening | Quantifier modifications or domain reductions lead to over-weak claims. |
| S1.3 | Logical Connective Misuse | Logical connectives are misused. |
| S2.1 | Object Type Error | Type of mathematical object are incorrectly stated. |
| S2.2 | Function Confusion | Function names are incorrectly selected. |
| S2.3 | Operator Confusion | Arithmetic operators are incorrectly selected. |
| S2.4 | Exponent/Power Error | Wrong value are assigned to polynomial degrees or exponential powers. |
| S2.5 | Coefficient/Constant Error | Wrong value are assigned to coefficients, multipliers, or constants in expressions. |
| S2.6 | Index/Subscript Error | Indices or subscripts of expressions are incorrect. |
| S2.7 | (Partial) Order Error | Relations of all kinds of orders are reversed, loosened, or tightened. |
| S3.1 | Infinity Misinterpretation | Statements about infinitude are misinterpreted (to wrong quantifier statement). |
| S3.2 | Extremum Concept Error | Extremum value are confused with bound. |
| S3.3 | Cardinality Error | Cardinality of a set is misrepresented |
| S3.4 | Integration/Diff. Confusion | Indefinite integration or differentiation is confused with definite integration. |
| S3.5 | Geometric Error | Wrong geometric objects/relations |
| **Constraint Error (C)** | | |
| C1.1 | Positivity Constraint Error | Positivity constraints are missing, redundant, or incorrect |
| C1.2 | Bound Constraint Error | Bound constraints other than positivity are missing, redundant, or incorrect |
| C1.3 | Domain Constraint Error | Wrong type/domain issues without algebraic structural changes |
| C1.4 | Variable Constraint Error | Other variable constraint errors |
| C2.1 | Range Shift | Wrong index range introduced by a constant offset shifting |
| C2.2 | Range Error | The size of the range changes, different from merely shifting |
| C3.1 | Missing Premise | Necessary assumptions are removed or absent |
| C3.2 | Redundant Premise | Unnecessary premises are added |
| C3.3 | Incorrect Premise | Wrong expression of an existing premise |
| C4 | Conclusion Error | Statement conclusion is incorrect |
| C5 | Auxiliary Construction Error | Error occur in helper definitions |
| **Implementation Error (I)** | | |
| I1 | Truncation Error | The truncation behaviors of discrete types lead to unexpected values |
| I2 | Operator Precedence Error | Missing/wrong parentheses lead to misassociation, altering the order of operations |

## F.2. SCI Definitions

**S1.1 Quantifier Strengthening:** Quantifiers are modified in a way that strengthens the statement. This includes: (1) Replacing existential quantifier $\exists$ with universal one $\forall$; (2) Expanding quantifier domain (e.g., 'first n terms' $\rightarrow$ 'all terms', finite domain $\rightarrow$ infinite domain).

**S1.2 Quantifier Weakening:** Quantifiers are modified in a way that weakens the statement. This includes: (1) Replacing universal quantifier $\forall$ with existential one $\exists$; (2) Reducing quantifier domain by changing 'all terms' to 'first n terms'.

**S1.3 Logical Connective Misuse:** Incorrectly use logical connectives: $\wedge, \vee, \rightarrow, \leftrightarrow, \neg, ....$ Equivalence transformation such as $\neg(a \vee b) \leftrightarrow (\neg a) \wedge (\neg b)$ are considered as correctly used.

**S2.1 Object Type Error:** Type selection that changes the mathematical object's essential properties or algebraic structure. This includes: (1) Basic algebraic structure changes (e.g., Bézout's theorem holds over $\mathbb{Z}$, but does not hold over the natural numbers, since $\mathbb{N}$ lacks additive inverses); (2) Changes on Group/Ring/Field or integral domain hierarchies (IntegralDomain, PID, UFD); (3) Other type changes that break theorems relying on specific algebraic properties.

Note: Basic range changes that only serves to limit the range of a variable without structural implications, should be classified as C1.3.

**S2.2 Function Confusion:** Incorrectly substitute one mathematical function for another, changing the mathematical operation being performed. This includes confusing trigonometric functions ($\sin \rightarrow \cos, \tan \rightarrow \cot, \arcsin \rightarrow \arccos$), number theoretic functions ($\gcd \rightarrow \text{lcm}, \text{Odd} \rightarrow \text{Even}$), rounding functions ($\text{ceil} \rightarrow \text{floor}, \text{round} \rightarrow \text{trunc}$), combinatorial functions ($\text{Combination} \rightarrow \text{Permutation}, C(n,k) \rightarrow P(n,k)$), logarithm bases ($ln \rightarrow log_{10}, log_2 \rightarrow log_{10}$), and other functions ($\text{abs} \rightarrow \text{sgn}, \min \rightarrow \max, \text{sqrt}\text{ß}\text{cbrt}$). Some functions have namespace variants (e.g., *Nat.log* vs *Real.log*). Changing between namespaces with different definitions also belongs here.

This category applies to mathematical functions only. It does NOT include operators, binary relations, exponents/powers, or coefficients.

**S2.3 Operator Confusion:** Arithmetic and algebraic operators are incorrectly substituted or modified, changing the mathematical operation. This includes confusing basic arithmetic operators , exponentiation and powers , root operations , operator precedence through parentheses changes , and modular arithmetic operators.

**S2.4 Exponent/Power Error:** Exponents or powers are incorrect, changing the mathematical function or expression structure. This includes polynomial degree errors, exponential sequence errors, and power errors in formulas.

**S2.5 Coefficient/Constant Error:** Numerical coefficients or constants in expressions are incorrect, changing which specific value or scaled version is being computed. This includes coefficients in function arguments, multipliers, additive constants, constants in final answers. Another case is: Units are incorrectly interpreted or converted. This includes using numerical values without proper unit conversion or applying incorrect transformation rules.

For example, directly using *45* to express 45 degrees instead of converting into $\pi/4$ radians. This kind of error is not limited to trigonometric functions.

**S2.6 Index/Subscript Error:** Indices or subscripts are incorrect, referencing a different element in sequences, arrays, or indexed collections. This includes sequence index errors and array element errors.

Note: If the function is range-related (*Finset.range*, *List.range*), classify the error under '**C2: Range Error**' instead. Only when the index in an array is wrong, it should be classified to S2.5. (e.g., *fun: n => n̂2* is changed to *fun: n => (n+1)̂2*, it doesn't matter with index, but the mapped value.)

**S2.7 Partial Order Error:** Statements about partial order relationship are misinterpreted. Typically a relation is reversed, loosened or tightened. Note that partial order relationship includes $|, \subseteq$, not limited to $<, \leq, >, \geq$.

**S3.1 Infinity Misinterpretation:** Statements about infinitude are misinterpreted. A statement claiming 'there are infinitely many numbers that satisfy property P' is not equal to 'any number n satisfies P'. Moreover, *{{a | P a}}.card* $= \infty$ is not equal to $\exists f: \mathbb{N} \rightarrow A, a\ n = f(n), P\ (a\ n)$. The latter statement is stronger as it claims A is countably infinite, while in the former statement A can be uncountable.

**S3.2 Extremum Concept Error:** Confusing 'bounded' with 'attaining extremum'. Problem asks for minimum/maximum value but formalization uses inequality which only shows a bound, not the actual extremum. Should use IsLeast/IsGreatest instead.

Key patterns: (1) IsLeast changed to incorrectly using $\geq$ (only states lower bound, not minimum); (2) IsGreatest changed to incorrectly using $\leq$ (only states upper bound, not maximum).

**S3.3 Cardinality Error:** Problem involves set cardinality (size/count, either finite or infinite) but formalization doesn't correctly express it. This includes all counting and set size problems.

Note that this error only relates to incorrectly used cardinality. The constant number modification such as changing *A.card=3* to *A.card=4* is not related with cardinality itself and should be classified to S2.4 Coefficient/Constant Error.

**S3.4 Integration/Differentiation Relationship Confusion:** Confusing the mathematical relationship between integration and differentiation when formalizing antiderivatives. In Lean, $\int$ notation produces a value (definite integral), not a function. To state 'F is an antiderivative of f', you must use *deriv F = f* (meaning F' = f), not $\int f = F$. There is no indefinite integral notation in Lean; the antiderivative relationship must be expressed through the derivative relation.

**S3.5 Geometric Relationship/Object Error:** Errors in selecting or expressing geometric objects, relationships, or constructions that alter the geometric meaning in pure/synthetic geometry contexts. This includes confusing different geometric entities or relationships.

For example, using the midpoint of diagonal AC instead of side AB, confusing an edge with a diagonal, switching geometric relationships, or referencing the wrong angle. This category applies to pure geometric reasoning only. It does NOT include algebraic/analytic geometry (coordinate-based calculations), trigonometric computations (sin/cos/tan), or geometric measurement type confusion (area vs perimeter).

**I1: Truncation Error:** This gap arises when operations or functions are applied to discrete numeric types, where Lean 4 uses truncated semantics that diverge from the mathematical reals. Three subcases are typical: (1) *Division Truncation*: division over discrete types is treated as truncated integer division rather than real-number division, so writing the expression as if over the reals may yield incorrect values; (2) *Subtraction Truncation*: natural number subtraction auto-truncates to zero, which is easy to overlook inside an absolute value or similar construct; (3) *Function Output Truncation*: some functions over discrete types return domain-specific values rather than their continuous counterparts.

**I2 Operator Precedence Error:** This gap is caused by missing or incorrect parentheses, leading to unintended operator precedence and altered expression semantics.

**C1.1 Positivity Constraint Error:** Errors related to positivity constraints on variables. This includes missing, redundant, or incorrect constraints that specify a variable must be positive.

**C1.2 Bound Constraint Error:** Errors related to bound constraints (upper/lower bounds other than positivity). This includes constraints like $n < 10, m \geq 6, k \leq 100$ that specify ranges or thresholds.

Examples: Missing $n \geq 6$ for Goldbach's conjecture; Redundant $n < 1000$ when not needed; Incorrect $n < 10$ changed to $n < 100$.

**C1.3 Domain Constraint Error:** Definitional domain errors where expressions are applied outside their valid domains. Type confusion without structural implications.

**C1.4 Variable Constraint Error:** Variable constraint errors not covered by C1.1 - C1.3. This is the catch-all for C1 category.

**C2.1 Range Shift:** The index range of a summation, product, or similar construction is shifted by adding or subtracting a constant, resulting in a different but superficially similar index set.

**C2.2 Range Error:** The range of indices is incorrectly specified, leading to missing, extra, or unintended elements. This includes off-by-one errors, incorrect interval boundaries, or replacing a finite range with an infinite one, while the overall structure remains well-typed.

**C3.1 Missing Premise:** Necessary assumptions or constraints are absent from the statement. A Lean theorem usually contains several hypotheses/assumptions. Focus on subtle but important assumptions that are missing.

For example: example {a b : Real} (h': a * 1/ b = 1) : a = b := sorry. You can identify that (h: b ≠ 0) is missing. Removing (h': a * 1/ b = 1) also creates misalignment, but it is too destructive.

**C3.2 Redundant Premise:** Unnecessary or incorrect premises are added to the statement. Additional constraints are introduced in ways not present in the original informal statement.

**C3.3 Incorrect Premise** Existing premises are incorrectly modified. Inequality constraints are weakened, strengthened, or altered in a way that deviates from the original semantic intent.

Note that a looser inequality is not wrong but it is still misaligned.

**C4: Conclusion Error:** The goal or conclusion of the statement is incorrect while premises may be correct. This is the ultimate catch-all category. Any misalignment that doesn't fit into S, I, or C1-C3 belongs here. If an error is related to what the theorem is trying to prove (rather than its assumptions), and doesn't fit other specific categories, it belongs to C4.

**C5 Auxiliary Construction Error:** Errors in auxiliary definitions, helper constructions, or local bindings that support the theorem but are neither the main theorem premises nor the conclusion.

These are supporting mathematical objects or statements that are incorrectly defined or constructed. This applies to function/constant definitions (*def*), helper lemmas (*lemma*), structure definitions (*structure*), inductive type definitions (*inductive*), and global/local variable definitions. For example, defining a helper function with wrong parameters or constructing an auxiliary lemma with incorrect bounds.

# G. Case analysis

## G.1. Representative Error Cases from Manual Review

---
**Case 1: S1.1 Quantifier Strengthening**

*Informal Statement:* Prove there exists $k \in \mathbb{N}^+$ such that for any $n \in \mathbb{N}^+$, $k \cdot 2^n + 1$ is always composite.
*Erroneous Formalization:*

```
example :
    ∀ k : ℕ, k > 0 ∧ ∀ n : ℕ, n > 0 → ¬ Nat.Prime (k * 2^n + 1) := by sorry
```

*Issue:* Universal quantifier ($\forall k$) instead of existential ($\exists k$). Claims ALL positive $k$ satisfy the property, not that at least ONE such $k$ exists.

---
**Case 2: S1.3 Logical Connective Misuse**

*Informal Statement:* If $2n - 1$ is prime, then for any $n$ distinct positive integers $\{a_1, a_2, \ldots, a_n\}$, there exist $i, j \in \{1, 2, \ldots, n\}$ such that $\frac{a_i + a_j}{\gcd(a_i, a_j)} \geq 2n - 1$.
*Erroneous Formalization:*

```
theorem entd1_2024_number_theory_12794
    (n : ℕ)
    (h_n_pos : 0 < n)
    (h_prime : Nat.Prime (2 * n - 1))
    (a : ℕ → ℕ)
    (a_unique : ∀ i ∈ Finset.Icc 1 n, ∀ j ∈ Finset.Icc 1 n, i ≠ j ∧ a i ≠ a j) :
    ∃ i ∈ Finset.Icc 1 n, ∃ j ∈ Finset.Icc 1 n, i ≠ j ∧ (a i + a j) / Nat.gcd (a
    i) (a j) ≥ 2 * n - 1 := by sorry
```

*Issue:* The *a_unique* condition uses conjunction ($\wedge$) instead of implication. Current form $i \neq j \wedge a\, i \neq a\, j$ is a conjunction that must hold simultaneously for all pairs, which is logically incorrect. Should use implication: $i \neq j \rightarrow a\, i \neq a\, j$ to express "if indices differ, then values differ".

---
**Case 3: S3.3 Cardinality Error**

*Informal Statement:* Does there exist infinitely many natural number sequences such that for every $n$, the sum $S_n$ is divisible by $2n^3$? Show that it is yes.
*Erroneous Formalization:*

```
example :
    ∃ a : ℕ+ → ℕ, ∀ n : ℕ+, 2 * (n : ℕ) ^ 3 |
    Σ i ∈ Finset.Icc 1 n, a i := by sorry
```

*Issue:* Proves existence of ONE sequence ($\exists\, a$), not infinitely many. The problem requires $|\{\text{solutions}\}| = \infty$, not merely nonemptiness of solution set.

---
**Case 4: C2.1 Range Shift**

*Informal Statement:* Select a subset $A$ from $\{1, 2, \ldots, 100\}$ such that no two consecutive integers are in $A$. How many ways are there? Show that it equals $\frac{1}{\sqrt{5}} \left[ \left( \frac{1+\sqrt{5}}{2} \right)^{102} - \left( \frac{1-\sqrt{5}}{2} \right)^{102} \right]$.
*Erroneous Formalization:*

```
example :
  Finset.card (Finset.filter (fun A : Finset ℕ => ∀ i ∈ A, i + 1 ∉ A)
  (Finset.powerset (Finset.range 100))) =
  (1 / Real.sqrt 5) * (((1 + Real.sqrt 5) / 2) ^ 102 -
  ((1 - Real.sqrt 5) / 2) ^ 102) := by sorry
```

> *Issue:* Finset.range 100 produces $[0, 100)$ instead of the required $[1, 100]$. Although it indexes 100 elements, the range is shifted: $\{0, 1, \ldots, 99\}$ rather than $\{1, 2, \ldots, 100\}$.

## G.2. Synthetic Negative Examples

### Example 1: S2.3 Operator Confusion

**Synthetic Negative Sample: Input**

*Informal Statement:* If $f$ is any function and $g$ is analytic without zero in $U$, then $f$ is meromorphic in normal form on $U$ iff $f \cdot g$ is meromorphic in normal form on $U$.
*Ground Truth Formalization:*

```
theorem meromorphicNFOn_mul_iff_left_of_analyticOnNhd {f g : \K → \K}
  (h₁g : AnalyticOnNhd \K g U) (h₂g : ∀ u ∈ U, g u ≠ 0) :
  MeromorphicNFOn (f * g) U ↔ MeromorphicNFOn f U := sorry
```

**Synthetic Negative Sample: Output**

*Synthetic Negative Sample:*

```
theorem meromorphicNFOn_mul_iff_left_of_analyticOnNhd {f g : \K → \K}
  (h₁g : AnalyticOnNhd \K g U) (h₂g : ∀ u ∈ U, g u ≠ 0) :
  MeromorphicNFOn (f + g) U ↔ MeromorphicNFOn f U := sorry
```

*Target Category:* S2.3 - Operator Confusion
*Localization: MeromorphicNFOn (f + g) U*
*Modification:* Changed multiplication operator * to addition operator + in the function composition.
*Reasoning:* The informal statement discusses multiplication of functions, which has special preservation properties when one factor is analytic and non-zero. Addition exhibits completely different behavior: the equivalence between $f$ being meromorphic and $f + g$ being meromorphic does not hold under the given hypotheses about $g$.

### Example 2: I2 Operator Precedence Error

**Synthetic Negative Sample: Input**

*Informal Statement:* Sophie Germain's identity: $a^4 + 4b^4 = (a^2 - 2ab + 2b^2)(a^2 + 2ab + 2b^2)$.
*Ground Truth Formalization:*

```
theorem pow_four_add_four_mul_pow_four' :
  a ^ 4 + 4 * b ^ 4 = (a ^ 2 − 2 * a * b + 2 * b ^ 2) *
  (a ^ 2 + 2 * a * b + 2 * b ^ 2) := sorry
```

**Synthetic Negative Sample: Output**

*Synthetic Negative Sample:*

```
theorem pow_four_add_four_mul_pow_four' :
  a ^ 4 + 4 * b ^ 4 = a ^ 2 − 2 * a * b + 2 * b ^ 2 *
  (a ^ 2 + 2 * a * b + 2 * b ^ 2) := sorry
```

*Target Category:* I2 - Operator Precedence Error
*Localization: a ^ 2 - 2 * a * b + 2 * b ^ 2 * (a ^ 2 + 2 * a * b + 2 * b ^ 2)*

*Modification:* Removed parentheses around the first factor *(a ^ 2 - 2 * a * b + 2 * b ^ 2)*, changing operator precedence.
*Reasoning:* Missing parentheses cause only *2 * b ^2* to multiply with the second factor, not the entire first polynomial. The expression now evaluates as $a^2 - 2ab + (2b^2)(a^2 + 2ab + 2b^2)$, producing a different expression.

### Example 3: C2.2 Range Error

Synthetic Negative Sample: Input

*Informal Statement:* Let the set $S = \{1, 2, \ldots, 50\}$. Find the smallest positive integer $n$ such that in every $n$-element subset of $S$, there are three numbers that can be the side lengths of a right triangle.
*Ground Truth Formalization:*

```
def S : Finset ℕ := Finset.Icc 1 50

theorem quantitative_reasoning_zh_blue_86 :
  IsLeast {n | 0 < n ∧ ∀ T ⊆ S, T.card = n → ∃ a ∈ T, ∃ b ∈ T, ∃ c ∈ T,
  a ≠ b ∧ a ≠ c ∧ b ≠ c ∧ a^2 + b^2 = c^2} 42 := by sorry
```

Synthetic Negative Sample: Output

*Synthetic Negative Sample:*

```
def S : Finset ℕ := Finset.Ico 1 50

theorem quantitative_reasoning_zh_blue_86 :
  IsLeast {n | 0 < n ∧ ∀ T ⊆ S, T.card = n → ∃ a ∈ T, ∃ b ∈ T, ∃ c ∈ T,
  a ≠ b ∧ a ≠ c ∧ b ≠ c ∧ a^2 + b^2 = c^2} 42 := by sorry
```

*Target Category:* C2.2 - Range Error
*Localization: Finset.Ico 1 50*
*Modification:* Changed *Finset.Icc* (closed interval $[1, 50]$) to *Finset.Ico* (half-open interval $[1, 50)$), excluding endpoint 50.
*Reasoning:* The interval type change produces the set $\{1, 2, \ldots, 49\}$ instead of $\{1, 2, \ldots, 50\}$. This alters the combinatorial problem since 50 could participate in Pythagorean triples, and the minimal $n$ value may differ for the reduced domain.

### G.3. Validation & Retagging Examples

We present two cases demonstrating the automated retagging process that ensures taxonomic accuracy.

### Example 1: Tag Correction (C4 → S1.3)

Initial Synthesis Output

*Informal Statement:* An equivalent definition for prime powers: $n$ is a prime power iff there is a prime $p$ and a natural $k$ such that $n$ can be written as $p^{k+1}$.
*Original Formalization:*

```
theorem isPrimePow_iff_pow_succ :
  IsPrimePow n ↔ exists (p : R) (k : N), Prime p /\ p ^ (k + 1) = n
```

*Modified Formalization:*

```
theorem isPrimePow_iff_pow_succ :
```

```
    IsPrimePow n → exists (p : R) (k : N), Prime p /\ p ^ (k + 1) = n
```

*Initial Synthesis Tag:* C4 - Conclusion Error

---

**Retagging Output**

*Corrected Tag:* S1.3 - Logical Connective Misuse
*Localization: IsPrimePow n -> exists (p : R) (k : N), ...*
*Reasoning:* The biconditional <-> was incorrectly changed to implication ->, weakening the equivalence statement to only one direction.
*Mapping Confidence:* High
*Mapping Notes:* This is a direct logical connective substitution ($\leftrightarrow \rightarrow \rightarrow$) which fits perfectly into the logical connective misuse category rather than the location-based C4.
*Correction Applied:* Initial tag C4 was too generic. The error's nature (logical connective substitution) takes priority over its location (conclusion), per the "Error Nature Over Location" principle. Retagged to S1.3.

**Example 2: Tag Confirmation (C2.2 $\rightarrow$ C2.2)**

**Initial Synthesis Output**

*Informal Statement:* Two solutions are equal iff they are equal on *range E.order*.
*Original Formalization:*

```
theorem sol_eq_of_eq_init (u v : N -> R)
   (hu : E.IsSolution u) (hv : E.IsSolution v) :
   u = v ↔ Set.EqOn u v (range E.order)
```

*Modified Formalization:*

```
theorem sol_eq_of_eq_init (u v : N -> R)
   (hu : E.IsSolution u) (hv : E.IsSolution v) :
   u = v ↔ Set.EqOn u v (range (E.order + 1))
```

*Initial Synthesis Tag:* C2.2 - Range Error

---

**Retagging Output**

*Validated Tag:* C2.2 - Range Error
*Localization: (range (E.order + 1))*
*Reasoning:* The range boundary is incorrectly extended by adding 1, changing the set of indices from $\{0, \ldots, \text{E.order} - 1\}$ to $\{0, \ldots, \text{E.order}\}$.
*Mapping Confidence:* High
*Mapping Notes:* This error directly fits C2.2 as it involves incorrect range specification with an off-by-one boundary error.
*Validation Result:* Initial tag confirmed correct. The modification is a prototypical range error with incorrect boundary specification.

### G.4. Compilation examples

We present two synthetic examples that illustrate the quality of our synthesized data. These examples also show that passing compilation checks alone is insufficient to guarantee semantic alignment.

**Example 1:** The synthesized data modified the statement's semantics without introducing any compilation error.

```
-- origin statement
```

```
theorem putnam_2020_a2 (k : ℕ): (Σ j ∈ Finset.Icc 0 k, 2 ^ (k - j) * Nat.choose (k +
    j) j = ((fun k ↦ 4 ^ k) : ℕ → ℕ ) k) := sorry

-- modified statement
theorem putnam_2020_a2 (k : ℕ): (Σ j ∈ Finset.range k, 2 ^ (k - j) * Nat.choose (k +
    j) j = ((fun k ↦ 4 ^ k) : ℕ → ℕ ) k) := sorry
```

**Example 2:** A complicated example synthesized from Mathlib data. Origin statement uses *hu_strict : StrictMono u*. The synthesized data is changed to *hu_strict : Monotone u*, without introducing any compilation error.

```
import Mathlib.Analysis.SpecialFunctions.Pow.NNReal
import Mathlib.Analysis.SpecialFunctions.Pow.Continuity
import Mathlib.Analysis.SumOverResidueClass

@[expose] public section
def SuccDiffBounded (C : ℕ) (u : ℕ → ℕ) : Prop :=   ∀ n : ℕ, u (n + 2) - u (n + 1) ≤
    C · (u (n + 1) - u n)
namespace NNReal

open Finset   open ENNReal in
theorem summable_schlomilch_iff {C : ℕ} {u : ℕ → ℕ} {f : ℕ → ℝ≥0}
    (hf : ∀ {m n}, 0 < m → m ≤ n → f n ≤ f m)
    (h_pos : ∀ n, 0 < u n)
    (hu_strict : Monotone u)
    (hC_nonzero : C ≠ 0)
    (h_succ_diff : SuccDiffBounded C u) :
    (Summable fun k : ℕ => (u (k + 1) - (u k : ℝ≥0)) * f (u k)) ↔ Summable f := sorry
```

# H. Test-Time Scaling via Self-Refinement

## H.1. Experimental Setting

We conduct a test-time scaling experiment using an iterative self-refine loop on a sample of 100 problems. In Round 0, Goedel-Formalizer-8B (Lin et al., 2025) generates 8 formalization candidates per problem, for a total of 800 candidates. In each subsequent round, a formal verifier identifies compiled but misaligned candidates, and its feedback is passed back to the formalizer to produce a new set of 8 refined candidates for those problems. We run up to 3 refinement rounds. This setup is related to concurrent work on iterative autoformalization (Yu et al., 2025a), though we focus specifically on the role of verifier feedback richness rather than the training pipeline.

We compare two verifier feedback conditions. FORMALRX provides rich structured feedback that includes the error type, the location of the error, and a corrected statement for reference; the formalizer receives a detailed description of what went wrong and is asked to reason carefully about the issue before producing a new candidate. LeanScorer (LS) provides a binary signal indicating only that the formalization was found to be incorrect by a formal alignment checker, with no further detail.

In each round, only problems that have at least one compiled but verifier-misaligned candidate are selected for refinement. Under FORMALRX this amounts to 47 problems in Round 1, 45 in Round 2, and 43 in Round 3. Under LS this amounts to 23 problems in Round 1, 23 in Round 2, and 27 in Round 3.

We report two pass metrics. Local Pass@8 is computed only over the subset of problems refined in that round, measuring the direct quality of the new candidates. Accumulated Pass@8 takes a best-of-rounds view over all 100 problems, marking a problem as passing if any round produced at least one DeepSeek-verified correct candidate. Accumulated Pass@8 is the primary metric for evaluating overall improvement across rounds.

## H.2. Main Results

Table 15 reports Pass@8 across self-refinement rounds for both verifier conditions; supporting statistics (compile rate, per-round candidate counts, verifier precision) appear in Table 16.

*Table 15.* Pass@8 across self-refinement rounds under structured (FORMALRX) and binary (LeanScorer) verifier feedback. Local Pass@8 (loc.) is computed over the refined subset of each round; Accumulated Pass@8 (acc.) is the best-of-rounds result over all 100 problems. Numbers in parentheses indicate the size of the refined subset evaluated in that round.

| Metric | R0 | FORMALRX R1 | FORMALRX R2 | FORMALRX R3 | LS R1 | LS R2 | LS R3 |
|---|---|---|---|---|---|---|---|
| Pass@8 (loc.) | 42.0% (100) | 46.8% (47) | 46.7% (45) | 41.9% (43) | 30.3% (33) | 34.8% (23) | 29.6% (27) |
| Pass@8 (acc.) | 42.0% | 46.0% | 49.0% | 49.0% | 42.0% | 43.0% | 44.0% |

## H.3. Analysis

**FORMALRX improves substantially and converges at Round 2.** Accumulated Pass@8 rises from 42.0% to 46.0% after Round 1 and reaches 49.0% after Round 2, then holds steady through Round 3. The net gain over baseline is 7 percentage points.

**LeanScorer improves slowly and achieves less.** Accumulated Pass@8 stays flat at 42.0% through Round 1, then rises to 43.0% and 44.0% in subsequent rounds. The net gain is only 2 percentage points after three rounds.

**The gap between FORMALRX and LeanScorer reveals the importance of feedback richness.** Both verifiers have comparable precision on Round 0 candidates (FORMALRX 46.3% vs LeanScorer 45.0% candidate-level). The difference in downstream improvement therefore stems from the quality of the feedback signal rather than verifier accuracy. A binary incorrect label gives the formalizer no information about what kind of error was made or where it occurred, making targeted correction effectively impossible. Structured feedback from FORMALRX allows the formalizer to understand and address the specific misalignment.

**LeanScorer local Pass@8 is far below the R0 baseline.** LeanScorer Round 1 local Pass@8 is 30.3% compared to R0's 42.0%, while FORMALRX Round 1 local Pass@8 is 46.8%, already above baseline. This confirms that FORMALRX is actively guiding the formalizer toward better solutions, whereas LeanScorer selects a hard subset and fails to meaningfully improve it.

**The Round 3 FORMALRX local regression is expected and benign.** Local Pass@8 drops to 41.9% in FORMALRX Round 3, below the R0 baseline. This is caused by verifier noise, where FORMALRX occasionally labels a correct formalization as misaligned, causing it to be unnecessarily refined and sometimes regressed in the process. Because the accumulated metric preserves each problem's best result across all rounds, overall Pass@8 remains stable at 49.0%. This is precisely why accumulated Pass@8 is the appropriate metric for multi-round self-refine evaluation.

## H.4. Supporting Statistics

Table 16 reports per-round compile rates, Pass@4 metrics, verifier precision, and the breakdown of gains versus losses against the Round 0 baseline.

*Table 16.* Supporting statistics across self-refinement rounds. *by cand* aggregates across all generated candidates; *by prob* aggregates per problem (a problem counts if any of its candidates qualifies). *loc.* and *acc.* denote the local and accumulated Pass@4 metrics.

| Metric | R0 | FORMALRX R1 | FORMALRX R2 | FORMALRX R3 | LS R1 | LS R2 | LS R3 |
|---|---|---|---|---|---|---|---|
| Compile Rate (by cand) | 51.4% | 95.7% | 96.7% | 98.5% | 95.8% | 97.3% | 97.2% |
| Compile Rate (by prob) | 75.0% | 100.0% | 100.0% | 100.0% | 100.0% | 100.0% | 100.0% |
| Pass@4 (loc.) | 35.0% | 44.7% | 42.2% | 39.5% | 30.3% | 26.1% | 29.6% |
| Pass@4 (acc.) | 35.0% | 40.0% | 41.0% | 41.0% | 36.0% | 36.0% | 37.0% |
| Verifier Prec. (by cand) | 46.3% | 26.3% | 36.9% | – | 32.9% | 29.0% | – |
| Verifier Prec. (by prob) | 60.0% | 37.9% | 44.0% | – | 31.2% | 35.0% | – |
| Problems refined | 47 | 45 | 43 | – | 23 | 23 | – |
| Gains vs R0 | 0 | 4 | 6 | 3 | 0 | 1 | 1 |
| Losses vs R0 | 0 | 2 | 2 | 2 | 5 | 2 | 3 |
| Net vs R0 | +0 | +2 | +4 | +1 | −5 | −1 | −2 |

**Compile rate** jumps from 51.4% at Round 0 to above 95% across all refinement rounds under both conditions. This confirms that the self-refine loop reliably produces syntactically valid Lean candidates, and that the quality improvement is semantic rather than structural.

**Verifier precision drops from Round 0 to Round 1** under both conditions. This is expected because the refined subset consists of problems that were already failing, where true alignment is rarer and verifier false positives are more frequent. FORMALRX precision recovers somewhat in Round 2 (36.9%) as the easiest-to-fix problems are resolved in Round 1.

**Gains and losses** show that FORMALRX achieves positive net gains in every round ($+2$, $+4$, $+1$ net), with consistent new passes (4, 6, 3) against modest losses (2, 2, 2). LeanScorer starts at net $-5$ in Round 1 with zero gains, recovering only partially in later rounds. The persistent losses under both conditions reflect verifier noise causing unnecessary refinement of problems that were already correctly formalized in earlier rounds.

# I. Prompts

## I.1. Training and Evaluation Prompt

---

### Diagnostic Annotation Prompt for Training and Evaluation

**Role:** You are an expert in evaluating alignment between informal mathematical problems and formal Lean4 statements.
**Task:**
- Determine if the formal statement correctly represents the informal problem
- If misaligned, identify the error type and location

**Output format (strictly follow):**
[ANSWER_BEGIN]
Semantic Alignment: [Aligned/Misaligned]
Error Type: [exact error type name from list, or N/A]
Error Location: [minimal code snippet, or N/A]
Corrected Statement:
[Complete corrected formal statement]
[ANSWER_END]

---

**Valid Error Types (MUST use exact names):**
*Semantic-Critical Error:* Quantifier Strengthening, Quantifier Weakening, Logical Connective Misuse, Object Type Error, Function Confusion, Coefficient/Constant Error, Partial Order Confusion, Operator Confusion, Index/Subscript Error, Exponent/Power Error, Infinity Misinterpretation, Extremum Concept Error, Cardinality Error, Unit Confusion, Integration/Differentiation Confusion, Geometric Relationship/Object Error
*Implementation Error:* Truncation Error, Operator Precedence Error
*Constraint Error:* Positivity Constraint Error, Bound Constraint Error, Domain Constraint Error, Variable Constraint Error, Range Shift, Range Error, Missing Premise, Redundant Premise, Incorrect Premise, Conclusion Error, Auxiliary Construction Error
**Critical Requirements:**
- Output ONLY exact error type name (e.g., "Quantifier Strengthening")
- DO NOT include category prefixes, IDs, or modifications
- If Aligned, use "N/A" for Error Type

---

**Output Field Guidelines:**
*Semantic Alignment:* "Aligned" if correct; "Misaligned" if error exist
*Error Type:* Exact name from list above, or "N/A" if aligned
*Error Location:* Minimal code snippet containing error; use premise section/goal section for missing items; "N/A" if aligned
*Corrected Statement:* If aligned, output original as-is; if misaligned, output complete corrected Lean4 statement

---

## I.2. Task 3 Judgment

---

### Localization LLM-as-Judge: Judgement of Error Localization

Compare two error location descriptions for the same formal statement
**Formal Statement:** {formal_statement}
**Predicted Error Location:** {predicted}
**Ground Truth Error Location:** {ground_truth}
Determine if both descriptions refer to the same error location.
**Special Cases:**
- If Ground Truth is "N/A": Score 1 if Predicted is also "N/A", else Score 0
- If either is "N/A" but not both: Score 0
**Score 1 (Same Location)** - Both point to the same buggy code:

---

- Same subterm with different wording: "$\forall : \mathbb{N}$" $\approx$ "universal quantifier" $\approx$ "quantifier $\forall$"
- Synonyms/abbreviations: "hypothesis h$\theta$" $\approx$ "premise h$\theta$" $\approx$ "h$\theta$"
- Different detail levels: "$\sum$" $\approx$ "$\sum$ i in range, f i "
- Wrapper vs content: "some $\langle 0, x \rangle$" $\approx$ "$\langle 0, x \rangle$" (outer constructor vs inner term)
- Parent vs child: "f (g x)" $\approx$ "g x" if the bug is in the argument

**Score 0 (Different Locations)** - Point to distinct error:
- Structurally separate: "conclusion" vs "premise"
- Different objects: "quantifier $\forall$" vs "summation $\sum$"
- Different values: "x$\hat{}$ 2" vs "x$\hat{}$ 3", "0" vs "1"
- Different components: "first equation" vs "second equation"

**Evaluation Principle:**
Score 1 if fixing the location in either description would fix the same bug.
Score 0 if they identify different bugs that require separate fixes.

**Response Format:**
[SCORE]<0 or 1>[/SCORE]
[REASON] < one line explanation> [/REASON]
Your response:

## I.3. Task 4 Judgment

### Correction LLM-as-Judge: Judgment of Equivalence of Statements

You are an expert in evaluating Lean4 statements. Determine if two formal statements are **semantically equivalent** (represent the same mathematical meaning).

**Informal Problem:** {informal_problem}
**Predicted Statement:** {predicted}
**Ground Truth Corrected Statement:** {ground_truth}

**Score 1 (Equivalent):** Same mathematical meaning despite differences in:
- Variable names, formatting, expression order
- Logically equivalent forms: $\neg(a \wedge b)$ vs $\neg a \vee \neg b$, $(\exists x.P) \vee (\exists x.Q)$ vs $\exists x.(P \vee Q)$
- Definitionally equivalent conditions in the type context (e.g., $\|v\|=0 \leftrightarrow v=0$ with normed spaces related environment open)
- Redundant constraints: explicit "i$\neq$j" when already implied by context
- Structural reformulation: different nesting of $\wedge, \vee, \rightarrow$, etc with same meaning

**Score 0 (Different):** Different mathematical claims like:
- Different quantifiers ($\forall$ vs $\exists$)
- Different domains ($\mathbb{N}$ vs $\mathbb{Z}$)
- Different operators or conclusions

**Key Principle:**
Determine if Predicted and Ground Truth are logically equivalent:
- Consider definitional properties of the types involved
- Do they have the same models (satisfying interpretations)?
- Does proving one automatically prove the other?
Use the informal problem to disambiguate intent when statements are close.
If Predicted and Ground Truth assert the same mathematical claim (even with different syntax), score 1.
Only score 0 if they make genuinely different claims.

**Response Format:**
[SCORE]<0 or 1>[/SCORE]
[REASON]<one line explanation>[/REASON]
Your response:

## I.4. Data Synthesis

### Negative Example Synthesis through SCI Taxonomy

**Role.** You are a formal verification expert specializing in Lean 4 theorem statements. Given a correct Lean statement paired with its informal version, generate negative examples by injecting specific misalignments drawn from the SCI taxonomy.
**Goal.** Produce statements that compile in Lean 4 but are semantically misaligned with the informal statement. The error must be realistic, the kind a human formalizer could plausibly write.

**Gap taxonomy.** Use the SCI gap definitions in Appendix F.2. Each gap describes a distinct way a formalization can drift from the informal statement.

**Category priority for ambiguous cases.** Apply S > I > C across categories. Within a single category, pick the gap with the greatest mathematical impact.

- **S (Semantic-Critical).** Changes to mathematical objects, structures, or concepts.
- **I (Implementation).** Formalization-specific computation or syntax issues.
- **C (Constraint).** Catch-all for conditions, ranges, premises, conclusions.

**Gap selection principle.** Pick the gap directly inferable from the difference between the informal statement and the formal code, not the downstream effect of that difference.

---

**Disambiguation cheat-sheet.** A few common ambiguities to handle consistently:

- **S1.1 vs S1.2 (quantifier direction).** S1.1 strengthens ($\exists \to \forall$, domain expansion); S1.2 weakens ($\forall \to \exists$, domain reduction). Extremum weakening goes to S3.2; cardinality weakening to S3.3.
- **S2.1 vs C1.3 (type error).** S2.1 covers algebraic structure keywords (Group, Ring, Field, IntegralDomain, PID, UFD). C1.3 covers basic carrier types ($\mathbb{N}, \mathbb{Z}, \mathbb{Q}, \mathbb{R}, \mathbb{C}$) without structural implications.
- **C1 subcategories.** C1.1 positivity ($> 0, \geq 1$); C1.2 other bounds ($n < 10, n \geq 6$); C1.3 type or domain; C1.4 catch-all.
- **S3.2 (extremum).** Problem asks for a minimum/maximum value. Use IsLeast/IsGreatest, not bare $\leq$ or $\geq$.
- **S3.3 (cardinality).** Problem asks about set size. Finite cases need .card; infinite cases need .Infinite.
- **C3 vs C4.** C3 is in assumptions; C4 is in the goal.

When multiple categories apply, pick the most specific and most central one.

---

**Modification rules.**

- **Theorem name.** Identical to the original, no suffix.
- **Compilation.** The modified statement must compile in Lean 4 with the same imports and header. If no compilable modification exists for a gap, output [GAP_NOT_APPLICABLE]; Reason: <brief explanation> for that gap.
- **Minimality.** Change only what the gap requires. Keep parameter order, naming, code style, and unrelated constraints identical.
- **Completeness.** Reproduce the entire formal statement, including every auxiliary structure, abbrev, def, and helper present in the original, even when unchanged. Output Lean code only, with no commentary or alternative variants.
- **Real difference.** The modified statement must differ from the original at the semantic level. Identical output, whitespace-only edits, and pure variable renames are not valid modifications.

**Permitted modification styles.** Either subtle (one operator or constraint changed) or structural (a piece of structure removed or replaced while compilation still holds). Both are acceptable as long as compilation succeeds and the result is genuinely misaligned.

---

**Diversity across modifications.** Across one problem's modifications, vary the transformation type. A good batch looks like:

- Modification 1: Finset.range $\to$ Set.Ico (constructor swap)
- Modification 2: drop a positivity constraint (constraint removal)
- Modification 3: flip a quantifier (logical structure change)

A bad batch (all the same trick) looks like:

- Modification 1: range 100 $\to$ range 101
- Modification 2: range 100 $\to$ range 99
- Modification 3: Icc 0 100 $\to$ Icc 0 101

**Avoid trivializing the problem.** Do not flip the math domain (e.g. geometry $\to$ number theory), remove every hypothesis, or otherwise disconnect from the informal statement.

---

**Header handling.** The input may include a header with imports, structures, abbrevs, and helper defs. Use it to resolve types and names while reasoning. Do not echo it in the output. The header is automatically prepended during compilation verification.

**Custom gap types.** If a realistic formalization error genuinely lies outside SCI, you may define a new gap as *Gap X - [name]* with a short description. Use this sparingly and only when no existing gap fits.

---

**Input.**

- **id:** {problem_id}
- **informal_statement:** {informal_statement}
- **header_section:** {header_section}
- **formal_statement (target to modify):** {formal_statement}

**Output.** For each applicable gap, return the complete modified Lean statement together with the gap label. One modification per gap. Output the Lean statements only, with no preamble or closing remarks.

## I.5. Validating Synthesized Data

### LLM as Judge: Retag as Verification

**Task:** Compare modified_formal with original_formal to identify what type of SCI error was introduced in modified_formal. Use informal_statement to understand the problem intent. I assume most original_target_gaps are reliable, but you still need to be careful.

**Input Data Fields**

Required:

- informal_statement: {informal_statement}
- original_formal: {original_statement}
- modified_formal: {modified_statement}
- original_target_gap: {original_target_gap}

**Special Task**

**Perform additional mapping analysis:**

After identifying the primary SCI gap using standard classification, add mapping evaluation:

1. Determine if this error can be reasonably mapped to the identified SCI category

2. Add three additional fields to your gap output:

- SCI_mapping: The SCI code you identified (e.g., "C4")
- mapping_confidence: "high", "medium", or "low"
- mapping_notes: Brief explanation (1-2 sentences)

**Mapping Confidence Levels:**

- high: The error clearly fits the identified SCI category without loss of meaning
- medium: Mappable but loses some specificity (e.g., "coefficient error" is better than "incorrect premise")
- low: Forcing into this category loses important semantic information

SCI Gap Classification Reference: [See F.2]

**Classification Guidelines**

**Fundamental Principle: Error Nature (WHAT) Over Error Location (WHERE)**

**Critical:** When classifying error, prioritize the nature of the error over its location in the code.

- The **type of error** (what mathematical/logical issue occurred) determines the classification - The **location** (premise, conclusion, auxiliary) is set as catch all categories

Use location-based categories (C3, C4, C5) only when no semantic/implementation category applies, or when the error is fundamentally about premise/conclusion structure itself.

**Three-way Comparison Process**

1. Read informal_statement to understand the problem intent 2. Compare modified_formal vs original_formal to identify differences 3. Determine error type based on what changed and how it affects the mathematical meaning

Key principle: The gap exists in modified_formal relative to original_formal, with informal_statement providing context for understanding intent.

**Primary Gap Principle**:

**When multiple modifications exist, report ONLY the primary gap** - the one with the greatest mathematical impact.

**Selection criteria:** 1. Which has the largest impact on mathematical meaning? 2. Which modification likely caused the others? 3. If fixing only one, which restores the most correctness?

**Gap Selection Principle**:

**Select the candidate gap type based on modification, instead of its potential effect**

**Category Priority:**[Same with I.4]

Your response:

