# OpenReview forum: "FormalRx: Rectify and eXamine Semantic Failures in Autoformalization"
_ICML.cc/2026/Conference — ICML 2026 regular_

### Official Review · Reviewer_b5u5 · 2026-03-11

**Soundness:** 3
**Presentation:** 1
**Significance:** 2
**Originality:** 3
**Overall Recommendation:** 4
**Confidence:** 4

**Summary:**

This paper introduces FORMALRX, a new evaluation framework for autoformalization based on a taxonomy of 28 distinct error categories with prioritized structure. To train an error diagnosis model, the authors construct a dataset by synthesizing misaligned pairs of formalizations with detailed diagnostic annotations. The resulting model, FORMALRX-8B, demonstrates stronger capability in diagnosing detailed errors in flawed formalizations compared to frontier LLMs.

**Compliance With Llm Reviewing Policy:**

Affirmed.

**Final Justification:**

Strengths include a well-organized multi-level error taxonomy, a reasonably detailed dataset construction pipeline, and a model that demonstrates practical ability to identify fine-grained error categories. These contribute positively to soundness and originality, particularly in structuring and operationalizing error diagnosis.

The main weakness remains clarity and positioning. The use of the term “evaluation framework” is not fully aligned with standard usage, which typically implies a quantitative metric for systematic comparison. FormalRx instead produces structured diagnostic outputs without a clearly defined scalar or aggregatable evaluation signal, limiting its interpretability as a benchmarking framework and weakening significance in the context of evaluation methodology.

The rebuttal partially improved clarity by better defining components, confirming model-agnostic design, and clarifying taxonomy construction. However, it did not fully resolve the conceptual ambiguity regarding evaluation versus diagnosis, which remains the central concern. Though authors agree to give a more precise description in the revised version.

Overall, I maintain my assessment but revise the score from 3 to 4, reflecting improved clarity after rebuttal while retaining concerns about positioning and evaluative framing.

**Key Questions For Authors:**

1. The description of the partition-based design in Section 3.1 appears to be conceptually inconsistent. The paper states that for any two distinct dimensions $D_1 \cap D_2=\emptyset$. However, the subsequent discussion mentions that “certain misalignments may exhibit characteristics matching multiple categories.” If this is the case, then the categories cannot be strictly disjoint, since some instances would naturally belong to multiple dimensions. While the authors introduce a priority assignment mechanism to resolve such overlaps, this does not fully address the issue that the taxonomy is described as a strict partition. Clarifying this design choice or revising the formulation of the taxonomy would improve conceptual consistency.

2. The paper states that the taxonomy was developed through an iterative cycle. Could the authors provide more details about the outcomes of each iteration? For example, how did the taxonomy evolve across iterations, and what criteria were used to determine convergence or stability of the taxonomy?

3. In Table 1, LeanScorer appears to underperform relative to several general-purpose LLMs. However, LeanScorer is designed specifically to provide alignment evaluation for formalizations. Could the authors explain why it performs worse than a relatively small general-purpose model such as Qwen3-4B with simple zero-shot prompting? Additionally, the results suggest that the Qwen models and frontier models show only limited differences in predicting the Verdict label. Could the authors comment on the reason for this? For example, could this be related to biases or artifacts in the curated dataset?

**Limitations:**

The authors could include a brief discussion of limitations, such as the potential lack of generalizability of FORMALRX to other mathematical domains and the reliance on curated datasets, which may introduce biases in error diagnosis.

**Strengths And Weaknesses:**

**Strengths**

1. The proposed three-tier taxonomy for formalization errors, covering semantic, constraint, and implementation levels, is well organized and clearly illustrated. The taxonomy is described in considerable detail, providing a structured framework for understanding different types of formalization errors.

2. The authors leverage the taxonomy to synthesize misaligned pairs of formalizations, resulting in a comprehensive dataset that could facilitate future research on formalization evaluation. This approach could also potentially serve as a data augmentation strategy for fine-tuning specialized models for autoformalization.

3. The resulting error diagnosis model, FORMALRX-8B, appears practically useful for identifying both error categories and detailed issues in formalizations, complementing the binary feedback typically provided by theorem provers.

**Weaknesses**

1. The organization of the paper could be improved. Although the motivation is discussed in the introduction, the paper’s contributions are not clearly articulated there. In particular, FORMALRX is described as an evaluation framework, but it is unclear whether it is model-specific or model-agnostic after reading the introduction. The abstract suggests that FORMALRX-8B is a model, which implies that the framework may be model-dependent, but this relationship is not clearly explained. Additionally, information about the constructed dataset is not presented in the introduction. These issues make it harder for readers to quickly understand the scope and contributions of the work.

2. The paper appears to lack comparisons with other taxonomy-based approaches for evaluating autoformalization. For example, the Epistemic Ensemble of LLM Judges (EFG) approach proposed in “Beyond Gold Standards: Epistemic Ensemble of LLM Judges for Formal Mathematical Reasoning” (Zhang et al., 2025) can also provide feedback for correcting formalizations and is applicable to both Lean4 and Isabelle. Without comparison or discussion of such approaches, the novelty and positioning of the proposed taxonomy may be less clear. In particular, it would be helpful to discuss how different taxonomies influence the evaluation of autoformalization.

3. While FORMALRX provides detailed diagnoses of formalization errors, the paper does not discuss how these diagnostic outputs can be aggregated into a scalar evaluation score. Such a metric would be important for quantitatively comparing different autoformalization methods. For example, how could FORMALRX be used to systematically compare model formalizations across systems such as specialized LLMs for formal reasoning outperforming general-purpose models? Whether the evalutaion using FORMALRX is consistent with established expectations is unclear.

Overall, the proposed FORMALRX framework appears useful for diagnosing errors in mathematical formalizations. However, the scope and positioning of the work could be clarified further to avoid confusion. In particular, FORMALRX seems primarily to function as an error diagnosis model, which supports qualitative analysis of formalization errors. Its role as a quantitative evaluation framework remains less clear and would benefit from further elaboration.

---

> ### Author Rebuttal · Authors · 2026-03-31
>
> Thank you for the insightful comments. We address each point below and include more illustrations in the anonymous repo: https://anonymous.4open.science/r/FormalRx_annoymized-0475/README.md
> ## re: W1 & W3.
> > FormalRx is described as an evaluation framework, but it is unclear whether it is model-specific or model-agnostic. Additionally, information about the constructed dataset is not presented in the introduction.
>
> FormalRx is a **model-agnostic evaluation framework**. It defines four diagnostic tasks grounded in the SCI Error Taxonomy, and any model can be evaluated within this framework, whether a fine-tuned specialist or a frontier LLM prompted zero-shot. This is reflected in our experiments, where GPT, DeepSeek, and Claude series models are all assessed under the same protocol. FormalRx-8B is one concrete instantiation of this framework, trained to perform all four tasks jointly in a single forward pass.
> >  Information about the constructed dataset is not presented in the introduction.
>
> We acknowledge this was not introduced in the introduction, have added in the revised [introduction](https://anonymous.4open.science/r/FormalRx_annoymized-0475/RevisedIntroduction.md). As described in *Section 4.2 Data Source* and Appendix E, we construct our dataset by combining verified positive samples with synthetically generated negatives annotated using the SCI taxonomy. The resulting test set, **FormalRx-Test**, will be open-sourced as the first semantic evaluation set with fine-grained diagnostic annotations together with our model weights.
> > The paper does not discuss how diagnostic outputs can be aggregated into a scalar evaluation score.
>
> We respectfully note that collapsing detailed diagnostic outputs into a single scalar score would sacrifice the interpretability that motivates our work. *FormalRx is designed to provide actionable diagnostic signal*, such as error type, location, and a corrected statement, which we believe is more informative for both human debugging and downstream improvement pipelines.
>
> ## re: W2.
> > The paper appears to lack comparisons with other taxonomy-based approaches, such as EFG (Zhang et al., 2025).
>
> Thank you for pointing out this related work. We will include a discussion of EFG in the revised manuscript to better position our contribution and highlight differences in evaluation granularity and interpretability.
>
> ## re: Q1
>
> We agree that raw error types are not mutually exclusive. We explicitly introduce a priority mechanism as part of the taxonomy definition. We define a deterministic procedure: *types are ordered by priority*, overlaps are resolved by assigning each instance to the highest-priority type. The partition property holds at the final categorization after disambiguation, rather than at the level of raw category definitions.
>
> ## re: Q2
> >Could the authors provide more details about each iteration.
>
> The iterative procedure contains: **(1) Draft.** We began with a coarse-grained taxonomy, similar to those generated by LLMs (ref. Q2, Reviewer 37Y4), consisting of loosely defined categories without a clear hierarchical structure. **(2) Diagnosis.** We then use LLM to classify under this taxonomy to obtain labels with confidence scores. By manually inspecting N samples across confidence levels, we identified systematic issues in the taxonomy design. **(3) Refinement.** We resolve issues and start a new iteration. Convergence is reached if **no modification is needed over 5 consecutive iterations**.
>
> The evolution across iterations contains: (1) introducing **priority** for category overlapping; (2) introducing a fallback category "Others", refined to C3-C5 for finer classification; (3) proposing the SCI hierarchy for structural clarity; (4) adding **overlooked but common misalignment patterns** such as indefinite integrals in Lean; (5) merging low-frequency categories (e.g., truncation errors consolidated into I1.1).
>
> To further validate practical usability, independent experts assign categories based on the taxonomy in our human study (Reviewer oXMU, Table 4), yielding Cohen's κ = 0.895. This confirms the taxonomy is both conceptually well-defined and operationally usable in practice.
> ## re: Q3.
> > Why does LeanScorer underperform general-purpose LLMs
>
> The performance gap for LeanScorer is not specific to our dataset. We observe the same trend on ConsistencyCheck and EPLA (see our response to Reviewer oXMU, W3), where LeanScorer consistently underperforms general-purpose models and even falls below BLEU on F1.
> > Could the limited differences between Qwen and frontier models be related to dataset bias?
>
> Regarding the limited differences between Qwen and frontier models on Verdict, T1 is a binary classification task that is relatively tractable for capable LLMs with zero-shot prompting. The same pattern holds on OOD datasets, suggesting this is not an artifact of our training distribution.

---

> > ### Author Rebuttal · Reviewer_b5u5 · 2026-03-31
> >
> > Thank the authors for their rebuttal. However, the terminology surrounding FormalRx remains unclear.
> >
> > The authors describe FormalRx as a model-agnostic evaluation framework, but this characterization is not sufficiently supported. An evaluation framework is generally expected to provide a mechanism for scoring or comparing different candidates, in this case, formalizations. If FormalRx is indeed an evaluation framework, it should enable ranking or systematic comparison across different autoformalization systems. However, the paper does not provide clear evidence that FormalRx can be applied in this way across different formalizations or model backends.
> >
> > Instead, FormalRx appears to focus on generating textual diagnostics for formalizations. If this is its primary functionality, it may be more accurately described as a new task or dataset aimed at producing explanations, rather than as an evaluation framework. As such, the current terminology may be misleading.
> >
> > Additionally, the description of FormalRx lacks consistency throughout the paper. After its introduction, it is not until Section 4 (Dataset, around line 254) that the authors clarify that “FormalRx maps the pair to a structured diagnostic sequence.” Soon after, components of FormalRx are described using terms such as “training instance” and “output sequence.” This presentation suggests that FormalRx is closer to a method for organizing or constructing training data, rather than a general-purpose evaluation framework.
> >
> > Given that these concerns remain insufficiently addressed in the rebuttal, I believe my original evaluation is still appropriate and will maintain my scores.

---

> > > ### Author Response · Authors · 2026-04-03
> > >
> > > Dear Reviewer,
> > >
> > > Thank you for your thoughtful acknowledgement and for clearly articulating the remaining concerns. Upon reflection, we recognize that the framing issues you identified are more substantive than we initially acknowledged in our rebuttal.
> > > >**Framing and organization of contributions.**
> > >
> > > We have revised the abstract and introduction to present FormalRx more precisely. Rather than positioning it as an abstract "evaluation framework," we now describe it as a **diagnostic package** comprising four concrete components: (1) the *SCI Error Taxonomy*, a hierarchical classification of 28 error categories; (2) *FormalRx-8B*, a diagnostic model trained on synthetically annotated data; (3) *FormalRx-Test*, a semantic evaluation benchmark with fine-grained annotations; and (4) a solution pipeline to address the semantic alignment problem in autoformalization.
> > >
> > > We believe this framing more honestly reflects the nature of the contribution and resolves the model-agnostic ambiguity.
> > >
> > > In the revised manuscript, we have restructured the [Abstract and Introduction](https://anonymous.4open.science/r/FormalRx_annoymized-0475/RevisedIntroduction.md) to establish this definition in the opening paragraph, ensuring readers understand FormalRx's nature before encountering its components in later sections.
> > > >**On scalar aggregation and the comparison with EFG**
> > >
> > > Having carefully read the EFG paper, we realized that your question on scalar aggregation and the comparison with EFG are in fact closely related, and would like to discuss them together.
> > >
> > > EFG is designed to **produce a scalar score correlated with human Likert-scale ratings**, serving as a ranking proxy for comparing autoformalization systems. Its taxonomy of 4 high-level aspects and 12 atomic properties is optimized for aggregation into a single interpretable score via linear ensemble. Rather than ranking formalizations, FormalRx **diagnoses specific failures by attributing errors to one of 28 fine-grained categories and localizing the problematic code segment**, which is a different goal.
> > >
> > > For use cases that require scalar comparison across systems, EFG's aggregation approach is well-suited; for use cases requiring actionable feedback on what went wrong and how to fix it, FormalRx's four task scores serve as a structured multi-dimensional profile—analogous to HELM-style evaluation rather than a single number. The two approaches are thus complementary, and *we plan to systematically investigate the downstream impact of these two different feedback paradigms in subsequent work*.
> > >
> > > We also note that EFG reports substantially lower correlation with human assessments on Lean 4 (0.479) than on Isabelle/HOL (0.662), reflecting the inherent difficulty of evaluating Lean 4 formalizations, which is precisely FormalRx is designed for. We will add a dedicated discussion of this complementary relationship in the related work section of the revised manuscript.
> > >
> > > ---
> > > Beyond the specific concerns above,  we would like to take this opportunity to revisit the value of FormalRx, which we view as a principled solution package to a genuinely underserved problem in autoformalization evaluation.
> > >
> > > >**1. On the evaluation landscape.**
> > >
> > > As we discussed with other reviewers, the two existing benchmarks for semantic alignment evaluation [EPLA and ConsistencyCheck], both have documented quality issues that lead to inconsistent model behavior (for instance, Sonnet-4 outperforms Sonnet-4.6 **by 14 pp** on EPLA). FormalRx-Test is motivated precisely by this instability, and we see it as a step toward more principled evaluation rather than a standalone contribution.
> > > >**2. On the necessity of fine-grained diagnostics over binary.**
> > >
> > > Binary feedback tells a formalizer that something is wrong, but not what or where. Our [test-time self-refinement experiment](https://anonymous.4open.science/r/FormalRx_annoymized-0475/TTS.md) directly quantifies this gap. Using FormalRx diagnostic outputs as verifier feedback improves autoformalization Pass@8 from 42.0% to 49.0% over three refinement rounds, compared to only 44.0% with binary feedback. Thhi provides empirical evidence that richer diagnostic signals translate into meaningfully better formalization quality.
> > > >**3. On the necessity of training a dedicated diagnostic model.**
> > >
> > > FormalRx demonstrates clear practical value even under these difficult conditions. Our [updated results](https://anonymous.4open.science/r/FormalRx_annoymized-0475/Main%20Result.md) show that by Progressive training method, FormalRx comprehensively outperforms currently the strongest frontier baseline Opus-4.6 across every diagnostic dimension. This directly answers the question of **whether a dedicated trained model is necessary**.
> > >
> > > ---
> > > We would sincerely thank you again for your time and thoughtful engagement, and hope the above clarifications and updated results have addressed your concerns and offer a clearer picture of what FormalRx is, why it is needed, and what it can do.

---

### Official Review · Reviewer_vCcU · 2026-03-12

**Soundness:** 3
**Presentation:** 3
**Significance:** 3
**Originality:** 3
**Overall Recommendation:** 5
**Confidence:** 3

**Summary:**

This paper introduces FormalRX, a diagnostic evaluation framework for autoformalisation which is the task of translating informal mathematical statements into formal code. Rather than just giving a binary aligned-misaligned verdict like all prior work, FormalRX provides four outputs: a verdict, an error category from a 28-class taxonomy (which is called SCI → Semantic, Constraint, Implementation), error localisation within the formal code, and a corrected statement. The authors build a fine-tuned 8B model trained on ~56k synthetically generated misaligned pairs. FormalRX-8B achieves F1 of 0.88 on verdict, 0.71 on categorisation, and accuracies of 0.75 and 0.73 on localisation and correction, outperforming frontier models like GPT-4.1 and Claude-Sonnet-4 on all tasks.

**Compliance With Llm Reviewing Policy:**

Affirmed.

**Final Justification:**

The authors addressed most of my concerns and I have adjusted my score to reflect this.

**Key Questions For Authors:**

1. The single-task vs multi-task ablation (Table 8) shows the multi-task model you actually deploy loses 13.3 points on correction accuracy compared to a dedicated correction model. Why not use single-task models in practice, or maybe report which configuration you recommend for deployment?

2. Can you provide evidence of synthesis quality at a larger scale, for example by reporting automatic re-tagging disagreement rates across all 28 categories rather than just overall accuracy?

3. In table 5 out-of-domain test DeepSeek-v3.2 outperforms FORMALRX-8B (F1 0.674 vs 0.625). This is a real-world naturalistically misaligned dataset, which is arguably the most important test case. Can you explain why the performance gap narrows so significantly compared to the in-domain test, and what this implies about how well the synthetic training distribution covers real errors?

4. Is there a plan to extend the framework beyond Lean 4? Which parts of the SCI taxonomy could transfer to Isabelle or Coq with minimal modification, and which would need complete redesign?

**Limitations:**

The paper does not include a limitations section, which should be added. Key limitations not discussed include: the entirely synthetic training data and the small manual validation coverage; the strong Lean 4 specificity; the multi-task vs single-task performance gap; and the lack of analysis of failure cases where FORMALRX produces a correction that compiles but is still semantically wrong.

**Strengths And Weaknesses:**

Soundess: The core methodology is solid. Using Lean's REPL to verify that synthetic negative examples actually compile is good, it separates semantic errors from syntax errors cleanly. The two-stage evaluation (exact match then LLM judge) makes sense for Tasks 3 and 4 since correct Lean statements can look very different syntactically. The human validation study is small but it works. However, there are some issues. So three quarters of the training data is synthetically generated, and the manual quality check covers only 200 of 56,673 samples which is only 0.35%. That's a thin basis for strong quality claims. More concerning is Table 8: single-task training beats the deployed multi-task model by 13 points on correction. The authors acknowledge this but ship the multi-task model anyway without a convincing justification. On the out-of-domain test (Table 5), DeepSeek-v3.2 outperforms FORMALRX-8B, which is the most realistic evaluation and quietly undermines some of the stronger claims. There's also a clear labeling inconsistency between Figure 2 and Table 14 for the S3 subcategories, which could just be a copy-paste error.

Presentation: Generally well written and easy to follow. Figure 1 does a good job motivating the problem. There's no dedicated limitations section and weaknesses are in the appendix, these should be in the main section. Also, "Unit Confusion" appears in the training prompt as a valid error type but isn't in the main taxonomy table, which would confuse anyone trying to reproduce this.

Significance: Paper focuses on an important problem, Binary alignment verdicts are a real bottleneck for autoformalization, and a system that tells you what went wrong, where, and how to fix it is genuinely useful for both human debugging and RLHF pipelines. The SCI taxonomy itself is a standalone contribution. The main limitation on significance is the Lean 4 specificity. The authors claim the taxonomy could transfer to Coq or Isabelle but provide no evidence of this.

Originality: Fine-tuned alignment models exist, LLM-as-judge evaluation exists, error taxonomies exist but building a principled exhaustive taxonomy for formal math, using it to drive synthetic data synthesis, and training a unified model to do all four diagnostic tasks in one pass hasn't been done before. The retagging pipeline as a data quality mechanism is a clean reusable idea. The honest finding that multi-task training hurts performance on the harder subtasks is also worth noticing.

---

> ### Author Rebuttal · Authors · 2026-03-31
>
> Thank you for the insightful comments. We address each point below and include more illustrations in the anonymous repo: https://anonymous.4open.science/r/FormalRx_annoymized-0475/README.md
>
> > **Q1 Why not use single-task models**
>
> Single-task models serve as a performance upper bound in our ablation, not a deployment target, FormalRx is designed as a unified diagnostic tool producing all four outputs in a single forward pass. Deploying four separate models would require four independent inference passes, quadrupling memory and latency in practice.
>
> More importantly, we have since identified a more effective training strategy. Progressive training introduces tasks incrementally across four stages, each initializing from the previous checkpoint and retaining all prior task data to prevent catastrophic forgetting. As a result, **progressive training surpasses single-task on Tasks 1 and 2, and substantially closes the gap on Tasks 3 and 4, achieving localization and correction accuracies of 0.778 and 0.792 (vs. 0.818 and 0.862 for single-task).** We also evaluated sequential training, which achieves competitive per-task performance in isolation but loses the ability to jointly produce all four diagnostic outputs after each stage. **A detailed analysis is provided [in this link](https://anonymous.4open.science/r/FormalRx_annoymized-0475/ProgressiveTraining.md).**
>
> > **Q2 Evidence of synthesis quality at a larger scale**
>
> We report the distribution across all categories of LLM confidence scores during re-tagging in [this link](https://anonymous.4open.science/r/FormalRx_annoymized-0475/ConfidenceDistribution.md). The results show that predictions are generally made with high confidence. A larger-scale expert evaluation that human annotators re-examined a substantially expanded subset of the data was conducted.
>
> Since the submission deadline, we recruited additional Lean experts with a plan to **scale our human validation to 1,000 samples, and have already completed 800**. Each expert evaluates 12 samples per task, with 2 samples per task overlapping to measure internal agreement. **The results further support the overall quality and consistency of our synthesis process.**
>
> |Task|Accuracy (%)|Agreement Rate (%)|Disagreements (n)|κ (avg)|
> |:-|:-:|:-:|:-:|:-:|
> |T1: Synthesis|78.3 (188/240)|84.2 (32/38)|6|0.684|
> |T2: Validation & Re-tag|86.2 (207/240)|94.7 (36/38)|2|0.895|
> |T3: Localization Judge|84.6 (203/240)|86.8 (33/38)|5|0.789|
> |T4: Correction Judge|89.6 (215/240)|92.1 (35/38)|3|0.842|
> |**Overall**|**84.7 (813/960)**|**89.5 (136/152)**|**16**|**0.803**|
>
> > **Q3 Performance on ConsistencyCheck**
>
> We appreciate this observation. However, ConsistencyCheck might not be the most representative test case for real-world errors. It also contains autoformalized result, and constructed based on miniF2F, whose quality has documented issues (Sec. 4.2), introducing noise into the evaluation signal itself.
>
> Furthermore, we conducted additional OOD evaluation on EPLA, and also extended our frontier model comparison to include more recent models. As reported in our response to Reviewer oXMU (W3), **FormalRx-1.7B achieves the global best F1 of 0.541 on EPLA, outperforming all frontier models including DeepSeek-v3.2, GPT series, and more recent models such as Claude Opus-4.6 and GPT-5.3-codex.** This suggests that the OOD picture is more nuanced than ConsistencyCheck alone implies.
>
> More broadly, we believe that a fine-grained, reliable semantic verifier like FormalRx opens up many promising directions, including using it to evaluate and improve formalizers in real deployment settings.
> In fact, we believe that real-word error distribution is closely related to 1)the domain you focus on, and 2)the formalizer used, which would better reflect by using semantic verifier like FormalRx during autoformalization.We are actively pursuing this as future work.
>
> > **Q4 Transfer to Isabelle or Coq**
>
> We have considered the transferability of the SCI taxonomy across different formal systems in our design. As discussed in Section 3 (Line 5), we believe the Implementation (I) dimension of our taxonomy is *closely tied to the language-specific features* of Lean 4, while the Semantic (S) and Constraint (C) dimensions capture more *general challenges* in formalization that are independent of a particular proof assistant. Based on this design, we anticipate that the S and C dimensions can transfer to other systems with only minimal modifications. In contrast, the I dimension would require more substantial adaptation. Since it encodes errors *related to Lean 4-specific constructs* (e.g., tactic behavior), extending it to other proof assistants would necessitate redesigning this part of the taxonomy to align with their respective language features and execution models.
>
> > **Limitation**
>
> We have added the [Limitation](https://anonymous.4open.science/r/FormalRx_annoymized-0475/Limitation.png) sections in the revised version

---

> > ### Author Rebuttal · Reviewer_vCcU · 2026-04-01
> >
> > I would like to thank the authors for their rebuttal! Many of my concerns have been addressed therefore, I have adjusted my scores accordingly!
> >
> > Remaining Questions:
> > 1. Can you share the full training details for progressive training (how many stages, what data at each stage) so others can reproduce it? Does progressive training also help on the out-of-domain test, or only in-domain?
> > 2. You introduce FormalRx-1.7B for the EPLA result but the rest of the paper uses 8B. Why switch models here? How does FormalRx-8B perform on EPLA?
> > 3. You criticize ConsistencyCheck's quality but still reported it as your main OOD benchmark in the paper. Why?
> > 4. For data quality - Can you share the per-category re-tagging confidence scores rather than just the overall distribution?
> >
> > Thank you.

---

### Official Review · Reviewer_oXMU · 2026-03-21

**Soundness:** 3
**Presentation:** 3
**Significance:** 3
**Originality:** 3
**Overall Recommendation:** 4
**Confidence:** 3

**Summary:**

The paper introduced FormalRx, which is an evaluation framework for autoformalization. The paper curated 28 hierarchical Semantic, Implementation, and Constraint (SCI) Error Taxonomy that was used to finetune an 8B model to provide granular diagnostics.

**Compliance With Llm Reviewing Policy:**

Affirmed.

**Key Questions For Authors:**

Please refer to the weaknesses.

**Strengths And Weaknesses:**

Strengths:
1. The paper is well written and easy to follow.
2. The paper identifies a crucial problem with existing frameworks that offer only scalar scores or binary verdicts. The FormalRx model can provide granular feedback like precise error locations and corrections.
3. The method has a priority ordering to prevent "lazy" categorization, which can help the judgment avoid overuse of catch-all buckets.
4. The empirical results are promising for in-domain tasks.

Weaknesses:
1. Even though the 28 SCI is designed to be pairwise disjoint and collectively exhaustive, it is extremely difficult to prove that they are rigorous and sufficient. The paper's framework is taxonomy-guided, they synthesize training errors by injecting SCI-defined categories, then refines those instances again with SCI labels. So the training and the validation are somewhat based on the taxonomy. It is hard to show that the SCI taxonomy is truly complete for long-tail failures.
2. The paper uses expert validation on 200 samples, which is relatively small compared to the whole dataset.
3. The model is trained almost on synthetical dataset, which may not fully reflect the true distribution of errors produced by humans or LLMs in practice.
4. Minor: Figure 2 has duplicate C5.

---

> ### Author Rebuttal · Authors · 2026-03-31
>
> Thank you for the insightful comments. We address each point below and include more illustrations in the anonymous repo: https://anonymous.4open.science/r/FormalRx_annoymized-0475/README.md
>
>
> > **W1 Hard to show that the SCI taxonomy is truly complete for long-tail failures**
>
> We agree that it is inherently challenging to prove completeness, especially for long-tail failures. We would clarify several points: (1) While theoretical exhaustiveness is hard to guarantee, our taxonomy is designed to be operationally complete. In particular, we introduce fallback categories (e.g., C3–C5) based on error location, ensuring that any error can still be consistently captured, even if it does not fit into a specific class. (2) Extremely rare error types are unlikely to support reliable fine-grained distinctions. In practice, low-frequency categories are either consolidated into higher-level categories or subsumed under existing ones. This allows the taxonomy to remain both expressive and robust, while avoiding over-fragmentation on long-tail cases.
>
>
> > **W2 Expert validation set is small**
>
> Since the submission deadline, we recruited additional Lean experts with a plan to scale our **human validation to 1,000 samples, and have already completed 800**. Each expert evaluates 12 samples per task, with 2 samples per task overlapping to measure internal agreement. The detailed setting follows Appendix C.1.1. The expanded results are reported in Table 2.
>
> *Table 2: Human study results validating LLM components in the diagnostic pipeline.*
> |Task|Accuracy (%)|Agreement Rate (%)|Disagreements (n)|κ (avg)|
> |:-|:-:|:-:|:-:|:-:|
> |T1: Synthesis|78.3 (188/240)|84.2 (32/38)|6|0.684|
> |T2: Validation & Re-tag|86.2 (207/240)|94.7 (36/38)|2|0.895|
> |T3: Localization Judge|84.6 (203/240)|86.8 (33/38)|5|0.789|
> |T4: Correction Judge|89.6 (215/240)|92.1 (35/38)|3|0.842|
> |**Overall**|**84.7 (813/960)**|**89.5 (136/152)**|**16**|**0.803**|
>
> > **W3 Synthetical dataset may not fully reflect the true distribution of errors produced in practice**
>
> We acknowledge this limitation, and will explicitly discuss it in the Limitations section of the final paper.
> We also carefully considered alternative data sources, found that:
> 1. **Existing datasets** [ConsistencyCheck, EPLA] provide no fine-grained error signal. EPLA contains no error type annotations at all, while ConsistencyCheck applies inconsistent labeling criteria across error types.
> 2. **Direct expert annotation**, without a principled guiding framework, faces fundamental issues:
>    - Experts may disagree on what constitutes a misalignment,
>    - Error type definitions cannot be unified across annotators,
>    - And it is not scalable. Collecting real misaligned pairs at training scale is even harder.
>
> Therefore we propose the SCI Error Taxonomy as the first comprehensive framework for systematically categorizing misalignments in the formalization process. Taxonomy-guided error injection then enables data construction that is both scalable and consistent under unified error definitions.
>
> To further evaluate the effectiveness and generalization of our framework, we conducted OOD evaluation on EPLA (1247 human-filtered samples). However, EPLA is constructed from miniF2F and Proofnet, whose ground-truth FL quality has documented issues (Sec.4.2), introducing noise into the evaluation signal itself. Moreover, EPLA exhibits a double distribution shift relative to our training data, in both label distribution and feature distribution.
>
> *Table 3: Out-of-domain evaluation on EPLA. **Bold-italic**: global best. *Italic*: group best.*
>
> Model|Accuracy|Precision|Recall|F1
> ---|---|---|---|---
> FormalRx-1.7B|0.662|0.470|0.638 |**0.541**
> **Open Source Model**|
> Qwen3-1.7B |0.563|0.361|0.519 |0.426
> **Frontier Model** |
> DeepSeek-v3.2|0.564|0.394|0.738|0.513
> GPT-4.1|0.728|0.629|0.314|0.419
> GPT-5-mini |0.743|0.627|0.432 |0.511
> GPT-5.2|0.633|0.438|0.625 |0.515
> GPT-5.3-codex|0.674|0.475|0.437|0.455
> Sonnet-4|0.754|0.661|0.432|*0.523*
> Sonnet-4.6|0.700|0.535|0.298|0.383
> Opus-4.6|0.755|0.668|0.424|0.519
> **Specialized Metrics**|
> LeanScorer|0.745|0.688|0.334|0.450
> BLEU (Best F1)|0.440|0.347|0.902|*0.501*
>
> Despite these challenges, FormalRx-1.7B achieves the global best F1 of 0.541, **outperforming all frontier models including DeepSeek-v3.2, GPT series, and Claude series**. This suggests that even under significant distribution shift, models trained on SCI-grounded synthetic data retain meaningful generalization to real misaligned pairs.
>
>
> > **W4 Figure 2 has duplicate.**
>
> We apologize for this oversight and have carefully proofread the entire manuscript and redrawn this [figure](https://anonymous.4open.science/r/FormalRx_annoymized-0475/Revised_SCI_taxonomy.png).

---

> > ### Author Rebuttal · Reviewer_oXMU · 2026-04-06
> >
> > I thank the authors' responses. I will maintain my positive scores.

---

### Official Review · Reviewer_37Y4 · 2026-03-22

**Soundness:** 3
**Presentation:** 3
**Significance:** 2
**Originality:** 2
**Overall Recommendation:** 4
**Confidence:** 3

**Summary:**

This paper proposes to improve autoformalization assessment through fine-grained diagnosis. They design SCI (Semantic, Constraint, Implementation) Error Taxonomy, a suite of 28 commonly encountered error categories, to classify failure modes. They further perform supervised tuning on synthetic data generated through LLM-based error injection based on the error taxonomy. Evaluating on four tasks (verdict, justification, localization, and correction) they show their fine-tuned model based on Qwen3-8B outperforms frontier models such as DeepSeek-v3.2, GPT-4.1, and Claude-Sonnet-4 on in-domain test set and also perform competitively on out-of-domain test set (ConsistencyCheck).

**Compliance With Llm Reviewing Policy:**

Affirmed.

**Final Justification:**

The authors' detailed rebuttal has addressed most of my concerns (especially, new discussions on the limitation of automatic error taxonomy generation and new test time scaling experiments), and hence I think this paper deserves a weak accept score. However, I do not recommend a higher score because I feel like the manual curation of error taxonomy in this paper is fundamentally not scalable, especially if we have new data with error patterns unseen from training. More exploration should spend on finding ways to incorporate some sort of automation in the error taxonomy classification pipline to improve robustness when new data is observed, even if full automation is not possible.

**Key Questions For Authors:**

1. I’m a bit confused about what “in-domain test set” in Table 3 consists of. Do the authors take their synthetically generated data and split it into train and test?
2. Have the authors tried automatically generate the error taxonomy instead of manual curation? What are challenges for automatic generation?
3. Can the authors use their fine-tuned model as a verifier to improve, for example, test-time scaling performance?

**Limitations:**

1. It seems like the authors didn’t discuss any limitations of their work in the paper (at least from the main text). Can the authors provide some discussions on the failure mode of the proposed method?
2. The authors only provide performance evaluation on one out-of-distribution dataset (ConsistencyCheck). And it seems like on this dataset, the performance advantage of their fine-tuned model does not hold consistently anymore (especially given the sharp improvement seen from the in-distribution experiments). This leads to doubts on the robustness of their method. I think the authors should evaluate on more out-of-distribution datasets to further evaluate the performance.
3. In Sec 5.2, the authors sometime fall back to an LLM-judge to evaluate the performance. I’m a little worried that for complicated tasks (especially task 4: correction), whether the LLM judge will be strong enough to detect equivalence. In the case of correction, will it make more sense to use a SMT solver to check whether the formulation is correct?

**Strengths And Weaknesses:**

**Soundness:**
- The error taxonomy proposed by the paper is technically sound. Comparing with previous works that only evaluate based on binary verdict, the error classification definitely provides more informative signal in case errors occur.
- However, I find the manual curation of the error taxonomy a bit non-robust. In Sec. 3.2 the authors say they need to begin by manually reviewing ~2000 statement pairs, followed by an iterative refinement procedure. This seems to be very labor intensive. Furthermore, if the data distribution changes and new error classes is required, it seems like a lot of effort need to be invested to update the error taxonomy again.
- The experiments is generally sound, but I have some questions regarding the in-distribution setup (see questions) and have some concerns regarding the limited out-of-distribution experiments (see limitations).

**Presentation:** The paper is overall well presented and easy-to-follow.

**Originality:** I find the error taxonomy in this work to be highly related to knowledge extraction/summarization works for the math domain, where there have been many literatures on extracting knowledge from LLM failure modes when solving math problems. In this sense, I find the idea of error taxonomy in this paper to be not that novel (especially the taxonomy is manually curated rather than automatically generated).

**Significance:** Based on my above discussion of soundness and originally, I think the paper definitely have some contribution to the autoformalization community, but I currently have doubts about the level of its significance.

---

> ### Author Rebuttal · Authors · 2026-03-31
>
> Thank you for the insightful comments. We address each point below and include more illustrations in the anonymous repo: https://anonymous.4open.science/r/FormalRx_annoymized-0475/README.md
>
> > **Originality**
>
> We acknowledge the related line of work, but note two key distinctions:
> 1. Autoformalization is a fundamentally different task from math, errors arise from the gap between informal language and formal statement, not addressed by prior taxonomy work. This is the first work that provides a fine-grained and structured signal for this domain.
> 2. Manual curation is not a weakness but a necessity. We refer the reviewer to our response below.(re:W2)
>
> > **Q1 In-domain test set**
>
> We thank the reviewer for pointing out this was not clearly described in the paper. As detailed in Appendix E.4, we construct our dataset by combining verified positive samples with synthetically generated negatives, split at the problem level with an 8:1:1 ratio for train/validation/test to prevent data leakage. The resulting test set, FormalRx-Test, will be open-sourced as the first semantic evaluation set with fine-grained diagnostic annotations together with our model.
>
> > **Q2 Automatically generating the error taxonomy**
>
> We attempted automatic taxonomy generation by prompting several strong models, including Opus 4.6, GPT-5.4, and DeepSeek V3.2, with explicit instructions to construct a complete, mutually exclusive, and collectively exhaustive taxonomy. We observed consistent limitations across all models (see [challenge analysis and examples](https://anonymous.4open.science/r/FormalRx_annoymized-0475/AutoTaxonomyGeneration/overview.md)):
> 1. **Lack of structural coherence.** Some models (e.g., Claude) attempted hierarchical grouping, but the resulting taxonomies lacked clear logical organization. Others (e.g., GPT-5.4, DeepSeek) produced unstructured lists without meaningful relationships between categories.
> 2. **Failure to ensure MECE.** None of the models enforced non-overlapping categories or introduced fallback categories to guarantee coverage, despite explicit instructions.
> 3. **Missing domain-critical errors.** None of the models identified certain prevalent error types found in existing datasets, such as S3.4, highlighting the gap between general LLM knowledge and domain-specific expertise.
> These findings motivate our choice of expert-driven, human-in-the-loop design.
>
> >**Q3 Used to improve test-time scaling**
>
> We conduct a test-time scaling experiment using an iterative self-refine loop. Starting from Round 0 (Goedel-8B, 100 problems from FormalRx-test), we apply up to 3 rounds of refinement using two feedback types: (1)  FRX: fine-grained feedback from FormalRx; (2) LS: binary feedback from LeanScorer.
>
> *Table 1: TTS results over 3 rounds of iterative self-refinement.*
> |Metric|R0|FRX R1|FRX R2|FRX R3|LS R1|LS R2|LS R3
> |:-|:-:|:-:|:-:|:-:|:-:|:-:|:-:|
> |Pass@8 (local)|42.0% (42/100)|46.8% (22/47)|46.7% (21/45)|41.9% (18/43)|30.3% (10/33)|34.8% (8/23)|29.6% (8/27)
> |Pass@8 (overall)|42.0%|46.0%|49.0%|49.0%|42.0%|43.0%|44.0%
>
> FRX feedback improves overall Pass@8 from 42.0% to 49.0% (+7pp, converging at Round 2), while LS reaches only 44.0% after 3 rounds (+2pp), confirming that rich structured feedback is essential for effective self-refinement ([detailed analysis](https://anonymous.4open.science/r/FormalRx_annoymized-0475/TTS.md)).
>
> >**L1 Limitations and discussions on the failure mode**
>
> Thanks for noticing, we have added the [Limitation](https://anonymous.4open.science/r/FormalRx_annoymized-0475/Limitation.png) sections in the revised manuscript. Additionally, Appendix G provides case analysis across all pipeline components, including G.1 *Representative Error Cases from Manual Review*. We will further expand this with additional failed cases to provide a more comprehensive discussion of failure modes.
>
> >**L2 Out-of-distribution evaluation**
>
> We refer the reviewer to our response to W3 above, where we report OOD evaluation on EPLA and discuss the challenges of obtaining reliable benchmarks in this setting. Additional human validation results are provided in our response to Reviewer oXMU(W2).
>
> > **L3 LLM judge and SMT solver**
>
> We agree that solver-based verification is a promising direction for ensuring correction quality. However, our current focus is on NL-FL semantic alignment, where correctness is defined relative to the informal statement rather than another formal reference, which limits the direct applicability of solver-based approaches. As for the reliability of our LLM judge, our expanded human validation study shows that Task 4 achieves an accuracy of 89.6% and an expert agreement rate of 92.1% (κ = 0.842). We refer the reviewer to our response to Reviewer oXMU(W2) for the full results.

---

> > ### Author Rebuttal · Reviewer_37Y4 · 2026-04-03
> >
> > Thank you for your detailed rebuttal. My concerns have been largely addressed, and hence I increased my score.

---

### Official Review · Reviewer_QEnR · 2026-04-01

**Soundness:** 2
**Presentation:** 3
**Significance:** 3
**Originality:** 4
**Overall Recommendation:** 3
**Confidence:** 4

**Summary:**

The paper introduces FORMALRX, a comprehensive evaluation framework for autoformalization (translating natural language math into formal code like Lean 4). Instead of standard binary verdicts (pass/fail) or scalar similarity scores (BLEU), the authors propose the SCI Error Taxonomy, which categorizes semantic errors into 28 distinct classes. They synthesized a dataset of over 56,000 misaligned pairs to train FORMALRX-8B, a model designed to output an alignment verdict, the specific error category, the localized buggy code, and a corrected formal statement.

**Compliance With Llm Reviewing Policy:**

Affirmed.

**Final Justification:**

The paper tackles a highly significant problem in formal mathematical reasoning: the inadequacy of surface-level evaluation metrics for autoformalization. The proposed SCI Error Taxonomy and the synthetic data generation pipeline are highly original and well-engineered. However, my primary concern lies with the evaluation of the metric itself.

By introducing both a novel categorical metric and a novel validation approach (relying purely on internal F1-scores and categorical human agreement), the paper creates a closed evaluation loop. It lacks a familiar, continuous anchor (such as a Spearman/Pearson correlation with human severity judgments) to objectively prove that the taxonomy captures the true severity of misalignments. Additionally, the paper currently lacks a dedicated limitations section addressing the inherent subjectivity and foundational assumptions of this constructed framework.

I am assigning an initial rating of Weak Reject. I am highly open to raising my score to an Accept if the authors can use the rebuttal period to provide a standard correlation baseline against continuous human judgments, and commit to adding an explicit limitations section regarding the subjective nature of their taxonomy.

**Key Questions For Authors:**

1. **Standardized Correlation:** Can you provide a standard correlation score to anchor your method? For example, mapping your 28 error types to a continuous severity scale and reporting a Spearman/Pearson correlation against human Likert-scale judgments of semantic misalignment. I am currently scoring this just below acceptance (a 4/10 in spirit, or Weak Reject), but I am highly willing to raise my score if you can validate your metric using a familiar, standard continuous baseline.
2. **Taxonomy Correctness vs. Clarity:** Your 82.5% inter-annotator agreement proves the taxonomy is readable/clear, but how do you objectively prove it is mathematically optimal without a standard external anchor?

**Limitations:**

The authors adequately discuss some limitations regarding data synthesis, but they fail to address the critical limitation of their own evaluation loop—specifically, the inherent risks of evaluating a novel categorical metric without anchoring it to established continuous human baselines.

P.S. The paper would greatly benefit from an explicit "Limitations" section. Specifically, the authors treat their novel evaluation framework as axiomatically superior to prior methods without providing sufficient justification. While hard empirical evidence of this superiority may be difficult to obtain, the authors must at least provide robust theoretical reasoning to support this assumption, rather than treating it as a self-evident truth.

tldr; you need a limitations section.

**Strengths And Weaknesses:**

**Strengths:**
* **Motivation:** Moving away from opaque, binary evaluations toward actionable, fine-grained diagnostics is a highly valuable direction for formal mathematics and LLM reasoning.
* **Data Generation:** The taxonomy-guided synthetic data generation pipeline is well-engineered and provides a novel way to train models on semantic code failures.
* Overall, the paper's principal domain is autoformalization evaluation, and it successfully highlights the critical limitations of current surface-level metrics.

**Weaknesses (Critical):**
* **Circular Evaluation (Double Innovation):** This paper proceeds to analyze a general theme of moving machine learning evaluation away from opaque, monolithic scores, but it attempts to innovate in too many directions at once. The authors propose a novel metric (the 28-class generative evaluator) AND a novel validation method (relying strictly on categorical human agreement/F1-scores instead of standard correlations).
* **Lack of Familiar Anchors:** By abandoning standard continuous baselines (like Pearson or Spearman correlations with human Likert-scale severity scores), it is incredibly difficult to verify if the framework is actually "correct." High inter-annotator agreement (82.5%) proves that the 28 definitions are clear and humans can follow them, but it does not mathematically prove that the taxonomy is optimal or captures the true severity of misalignment. Without an objective, standard anchor to evaluate the evaluator, the paper asks the reader to take a massive leap of faith.

---

### Decision · Program_Chairs · 2026-04-30

**Decision:**

Accept (regular)

**Comment:**

This paper introduces FormalRx, a structured approach for diagnosing errors in autoformalization, including a fine-grained taxonomy, a synthetic dataset, and a trained diagnostic model. Overall, reviewers find the problem important and underexplored, and appreciate the practical value of moving beyond binary feedback toward more detailed error analysis.

Several concerns were raised. One reviewer questioned the novelty and positioning relative to prior work, while another raised concerns about evaluation setup and generalization. A key recurring issue is the framing of FormalRx as an “evaluation framework,” given that it does not provide a scalar metric for model comparison but instead produces structured diagnostic outputs. While the rebuttal clarified parts of this design, some ambiguity remains.

That said, multiple reviewers agree the method is technically sound and empirically useful, and the concerns are mainly about framing and scope rather than core validity. Overall, I lean toward acceptance.